# INSTRUCTPIX2NERF: INSTRUCTED 3D PORTRAIT EDITING FROM A SINGLE IMAGE

**Jianhui Li[1], Shilong Liu[1], Zidong Liu[1], Yikai Wang[1], Kaiwen Zheng[1], Jinghui Xu[2]**
**Jianmin Li[1]\*, Jun Zhu[1,2]\***
[1]Dept. of Comp. Sci. & Tech., Institute for AI, BNRist Center, THBI Lab, Tsinghua-Bosch
Joint ML Center, Tsinghua University, Beijing, 100084 China
[2]Shengshu Technology, Beijing
`{lijianhu21,liusl20,liu-zd20}@mails.tsinghua.edu.cn`
`yikaiw@outlook.com, zkwthu@gmail.com, jinghui.xu@shengshu.ai`
`{lijianmin,dcszj}@tsinghua.edu.cn`

## ABSTRACT

With the success of Neural Radiance Field (NeRF) in 3D-aware portrait editing, a variety of works have achieved promising results regarding both quality and 3D consistency. However, these methods heavily rely on per-prompt optimization when handling natural language as editing instructions. Due to the lack of labeled human face 3D datasets and effective architectures, the area of human-instructed 3D-aware editing for open-world portraits in an end-to-end manner remains under-explored. To solve this problem, we propose an end-to-end diffusion-based framework termed **InstructPix2NeRF**, which enables instructed 3D-aware portrait editing from a single open-world image with human instructions. At its core lies a conditional latent 3D diffusion process that lifts 2D editing to 3D space by learning the correlation between the paired images' difference and the instructions via triplet data. With the help of our proposed token position randomization strategy, we could even achieve multi-semantic editing through one single pass with the portrait identity well-preserved. Besides, we further propose an identity consistency module that directly modulates the extracted identity signals into our diffusion process, which increases the multi-view 3D identity consistency. Extensive experiments verify the effectiveness of our method and show its superiority against strong baselines quantitatively and qualitatively. Source code and pretrained models can be found on our project page: `https://mybabyyh.github.io/InstructPix2NeRF`.

## 1 INTRODUCTION

While existing 3D portrait editing methods (Cai et al., 2022; Lin et al., 2022; Sun et al., 2022b; Li et al., 2023; Xie et al., 2023; Lan et al., 2023) explored the latent space manipulation of 3D GAN models and made significant progress, they only support preset attribute editing and cannot handle natural language. The explosion of language models has made it possible to enjoy the freedom and friendliness of the natural language interface. Recently, many excellent text-supported image editing methods have emerged, such as Talk-To-Edit (Jiang et al., 2021), StyleCLIP (Patashnik et al., 2021), TediGAN (Xia et al., 2021), AnyFace (Sun et al., 2022a) and InstructPix2Pix (Brooks et al., 2023). However, these methods are typically limited to the 2D domain and cannot directly produce 3D results. While it is possible to connect a 2D text-supported editing model to a 3D inversion model for text-supported 3D-aware editing, this will result in extra loss of identity information as the original face is invisible in the second stage. Thus, an end-to-end model is more desirable for better efficiency and performance.

Rodin (Wang et al., 2023) and ClipFace (Aneja et al., 2023) explored end-to-end text-guided 3D-aware face editing. Rodin trains a conditional diffusion model in the roll-out tri-plane feature space

---
*Corresponding authors.

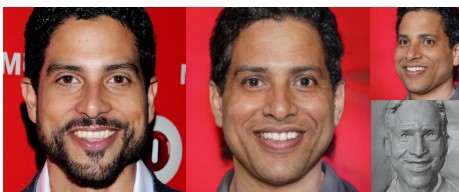

Remove the beard

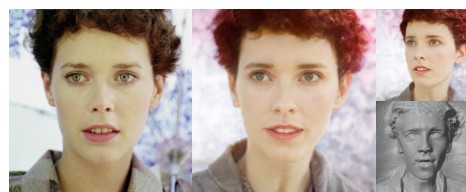

Turn the hair color to red

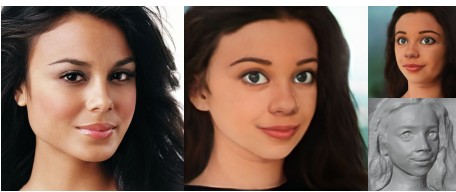

Make her look like a cartoon

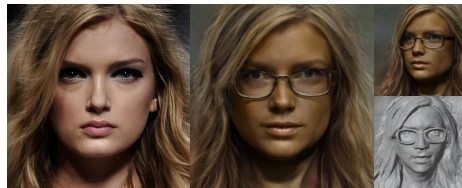

Put eyeglasses on her and turn the portrait into a bronze statue

Figure 1: Our instructed 3D-aware portrait editing model allows users to perform interactive global and local editing with human instructions. This can be a single attribute editing or style editing instruction, multiple attribute editing instruction together, or even attribute and style instructions together.

with 100K 3D avatars generated by a synthetic engine. Rodin achieves text-guided 3D-aware manipulation by conditioning the diffusion process with the manipulated embedding, which consists of the CLIP (Radford et al., 2021) image embedding and a direction in the CLIP text embedding. However, since Rodin can only generate avatar faces, it cannot handle real-world face editing. Clip-Face learns a texture mapper and an expression mapper with CLIP loss and uses the generated UV map and deformed 3D mesh to achieve text-guided manipulation in the synthetic domain. However, ClipFace cannot handle real-world face editing and is time-consuming and inconvenient because a separate encoder needs to be trained for each text prompt.

Human instructions are better for expressing editing intent than descriptive prompts. However, focusing on noun phrases, existing text-guided 3D-aware face editing methods are challenging to understand verbs in instructions. For example, descriptive-prompt-driven editing methods usually treat "*remove the eyeglasses*" as putting on the eyeglasses, as shown in Appendix A.2.1. Therefore, end-to-end 3D-aware portrait editing from an input image with human instructions is a critical and fascinating task, which aims to achieve 3D-consistent editing with a user-friendly interface. To our knowledge, we are the first to explore this area. We analyze that it is critical to design an efficient framework incorporating human instructions with 3D-aware portrait editing. In addition, due to the lack of multi-view supervision, it is challenging to maintain the consistency of identity and editing effects when performing 3D-aware editing from a single image, especially multi-semantic editing.

In this work, we address these issues by proposing a novel framework termed **InstructPix2NeRF**, which enables precise 3D-aware portrait editing from a single image guided by human instructions. We prepare a triplet dataset of each sample consisting of a 2D original face, a 2D edited face, and a human instruction semantically representing a single change from the original face to the edited face. Training on a large number of images and single human instructions, our model enables instructed 3D-aware editing with many single instructions even multiple instructions, rather than training the model for each human instruction. Although trained on a single-instruction dataset, InstructPix2NeRF enables multi-instruction editing during inference. InstructPix2NeRF can perform instructed 3D-aware face editing thanks to the three key ingredients below.

Firstly, we design a novel end-to-end framework combining diffusion and NeRF-based generators. The state-of-the-art architecture for text-guided image editing is typically based on diffusion models (Rombach et al., 2022; Zhang & Agrawala, 2023; Brooks et al., 2023). At the same time, 3D portrait modeling commonly relies on NeRF-based GANs (Chan et al., 2022; Sun et al., 2022b; Niemeyer & Geiger, 2021; Chan et al., 2021). We efficiently combine these two aspects to design an effective cross-modal editing model. Specifically, we use the inversion encoder PREIM3D (Li et al., 2023) to obtain the $w+$ latent codes of the 2D original and edited faces, then train a transformer-based diffusion model in the 3D latent space. The CLIP's text embedding of instructions is injected into

the diffusion process via a cross-attention block after the self-attention block. Latent codes with biconditional sampling of text and images are decoded to geometry and multiple views with a single pass through the NeRF-based generator.

Secondly, we propose a Token Position Randomization (TPR) training strategy to handle multiple editing requirements via one single pass. By TPR, instruction tokens are put in a random position of the token sequence, enabling the model to fulfill multiple editing requirements simultaneously while preserving the facial identity to a large extent.

Thirdly, we propose an identity consistency module that consists of an identity modulation part and an identity regularization part. We replace layer-normalization layers in the transformer block with adaptive layer norm (adaLN), which modulates the identity signal. The identity regularization loss is calculated between the original face image and a typical face image generated by a one-step prediction of latent code when the diffusion timestep is less than the threshold.

Combining the three key components together, InstructPix2NeRF enables instructed and 3D consistent portrait editing from a single image. With one 15-step DDIM(Song et al., 2021; Lu et al., 2022) sampling, our model can output a portrait and geometry with attributes or style guided by instructions in a few seconds. Figure 1 shows the editing results produced by our method. We recommend watching our video containing a live interactive instruction editing demonstration. To facilitate progress in the field, we will be completely open-sourcing the model, training code, and the data we have curated. We expect our method to become a strong baseline for future works towards instructed 3D-aware face editing.

## 2 RELATED WORK

**NeRF-based 3D generation and manipulation**. Neural Radiance Field (NeRF) (Mildenhall et al., 2021) has significantly impacted 3D modeling. Early NeRF models struggle to generate diverse scenes since the models are trained on a lot of pose images for every scene. Recent NeRF methods, such as GIRAFFE(Niemeyer & Geiger, 2021), EG3D(Chan et al., 2022), and IDE-3D(Sun et al., 2022b), integrate NeRF into the GAN framework to generate class-specific diverse scenes. These works have paved the way for 3D-aware objects editing methods like IDE-3D (Sun et al., 2022b) PREIM3D (Li et al., 2023), HFGI3D (Xie et al., 2023), and E3DGE (Lan et al., 2023) that perform semantic manipulation in the latent space of NeRF-based GANs in specific domain, such as human faces, cars, and cats. Despite the promising results of these methods, they cannot handle natural language.

**Diffusion models**. The diffusion probabilistic models (DPMs) (Sohl-Dickstein et al., 2015; Song & Ermon, 2019; Ho et al., 2020) can generate high-quality data from Gaussian noise through a forward noise addition process and a learnable reverse denoising process. By training a noise predictor using UNet (Ronneberger et al., 2015; Dhariwal & Nichol, 2021; Song & Ermon, 2019; Bao et al., 2022b) or transformers (Vaswani et al., 2017; Bao et al., 2022a; Peebles & Xie, 2022) backbone, the diffusion model can stably learn the probability distribution on ultra-large datasets and hold extensive applications such as image generation (Rombach et al., 2022), multi-model data generation (Bao et al., 2023), image editing (Zhang & Agrawala, 2023; Brooks et al., 2023), and likelihood estimation(Zheng et al., 2023). Initially, the diffusion model had difficulty generating high-resolution images in pixel space. Recently, the Latent Diffusion (Rombach et al., 2022), which combines the

Table 1: An overview of 3D-aware portrait editing methods.

| Method | Per-image Optimization-free | Real-world Images | Text-supported | Per-text Optimization-free | Instructed |
|---|---|---|---|---|---|
| IDE-3D | ✗ | ✓ | ✗ | - | ✗ |
| E3DGE | ✓ | ✓ | ✗ | - | ✗ |
| PREIM3D | ✓ | ✓ | ✗ | - | ✗ |
| ClipFace | ✗ | ✗ | ✓ | ✗ | ✗ |
| IDE3D-NADA | ✗ | ✓ | ✓ | ✗ | ✗ |
| Rodin | ✓ | ✗ | ✓ | ✓ | ✗ |
| Ours | ✓ | ✓ | ✓ | ✓ | ✓ |

VAE and the diffusion model, significantly improves generation efficiency and image resolution. Inspired by latent diffusion (Rombach et al., 2022), we combine the diffusion model with NeRF-base GAN to efficiently implement latent 3D diffusion.

**Text-guided editing**. The previous approach for text-guided editing (Jiang et al., 2021) usually involves training a text encoder to map text input or human instructions to a linear or non-linear editing space, which may not handle out-of-domain text and instructions well. Recently, The pre-trained CLIP (Radford et al., 2021) model contains vast vision and language prior knowledge, significantly accelerating the development of vision-language tasks. StyleCLIP (Patashnik et al., 2021) calculates the normalized difference between CLIP text embeddings of the target attribute and the neutral class as the target direction $\Delta t$, which is then applied to fit a style space manipulation direction $\Delta s$. InstructPix2Pix combines the abilities of GPT (Brown et al., 2020) and Stable Diffusion (Rombach et al., 2022) to generate a multi-modal dataset and fine-tunes Stable Diffusion to achieve instructed diverse editing. However, these methods are focused on editing 2D images and do not enable 3D-aware editing. Recently, Rodin (Wang et al., 2023), ClipFace (Aneja et al., 2023), and IDE3D-NADA (Sun et al., 2022b; Gal et al., 2022) explored text-guided 3D-aware editing. However, Rodin and ClipFace can only be applied to the synthesis face, and ClipFace and IDE3D-NADA require optimization for each text prompt. Meanwhile, Instruct-NeRF2NeRF (Haque et al., 2023), AvatarStudio (Mendiratta et al., 2023), and HeadSculpt (Han et al., 2023) have achieved success in optimizing text-driven single-3D-scene editing. However, these methods require a 3D scene rather than a single image, and they take tens of minutes for each scene.

As shown in Table 1, IDE-3D, E3DGE, and PREIM3D cannot handle natural language, ClipFace and IDE3D-NADA rely on per-prompt optimization, Rodin and ClipFace cannot be applied to real-world faces.

## 3 DATA PREPARATION

Inspired by InstructPix2Pix (Brooks et al., 2023), we prepared a multimodal instruction-following triplet dataset, where triplet data consists of an original face, an edited face, and a human instruction representing a single change from the original to the edited. Specifically, given a face, we use pretrained 2D editing models to produce a 2D edited face and use the large language model ChatGPT (Brown et al., 2020) to generate the corresponding editing instructions.

**Paired image generation.** We leverage two off-the-shelf 2D image editing methods, e4e (Tov et al., 2021) and InstructPix2Pix (Brooks et al., 2023), to generate paired images on FFHQ (Karras et al., 2019). We use e4e, the widely used state-of-the-art face attribute editing method, to generate 22K pairs of images with 44 attributes editing. As for InstructPix2Pix, we have selected 150 portrait-related instructions from its instruction set using keywords. The faces edited with these instructions were filtered by a face identification model (Deng et al., 2021) to obtain 18K paired images.

**Instruction generation.** In-context learning in large language models (LMs) demonstrates the power that rivals some supervised learning. For the paired images generated by e4e, we provide ChatGPT with some example data consisting of the paired attribute labels and corresponding hand-written transfer instructions. We tell it that the instructions represent a single change between attributes. After that, we asked ChatGPT to generate 20-30 transfer instructions with the provided attributes of the pairs. For the paired image generated by InstructPix2Pix, since there are already human instructions, we asked ChatGPT to generate 20-30 instructions representing the same semantic meaning on this basis.

Assigning human instructions to 40K paired images, we obtained a final triplet dataset containing 640K examples, each consisting of a single semantic instruction and paired image. See Appendix for more detailed data preparation.

## 4 METHOD

Given input image and human instructions, our goal is to generate multi-view images and geometry, which behave as intended by human instructions and can keep other attributes and identities unchanged. With the NeRF-based generator and inversion encoder, our goal can be translated into generating latent code $w$ that represents the edited face image. The latent code $w$ will be given to the

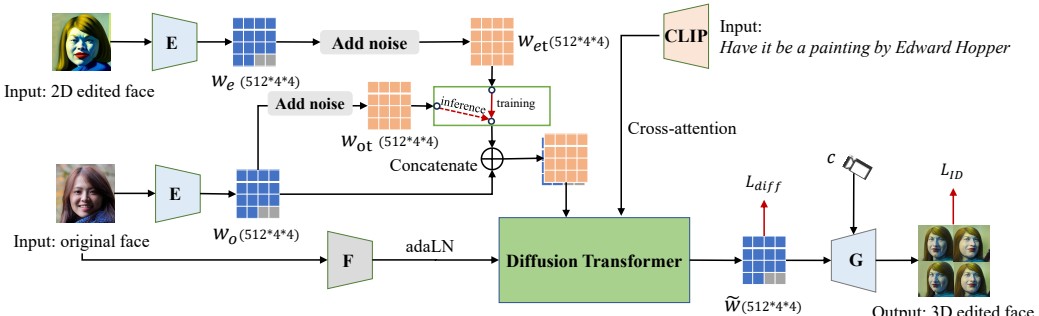

Figure 2: An overview of our conditional latent 3D diffusion model. We use a NeRF-based generator inversion encoder $E$ to obtain the latent code of the image. Then, we train a diffusion model conditioned on human instructions and the original face. Text instruction conditioning is introduced using the cross-attention mechanism with the CLIP text embedding, and the original image conditioning is realized via concatenating and adaptive layer norm. $F$ is a face identification model. $G$ is a NeRF-based generator. The Diffusion Transformer is trainable, the other models are fixed.

NeRF-based generator to produce multi-view images conditioned on camera pose. Figure 2 illustrates the whole architecture of our method. We will introduce the key components of our pipeline in the following subsections.

## 4.1 CONDITIONAL LATENT 3D DIFFUSION

Our method is based on the NeRF-based generator $G$ and inversion encoder $E$. The NeRF-based generator, such as EG3D (Chan et al., 2022), can generate multi-view images from a Gaussian noise $z \in \mathcal{Z} \subseteq \mathbb{R}^{512}$ conditioned on camera parameters $c$. The noise $z$ is mapped to the intermediate latent code $w = f(z) \in \mathcal{W} \subseteq \mathbb{R}^{k*512}$, which is used to produce tri-plane features through the convolutional network. A small MLP decoder is used to interpret the features retrieved from 3D positions as color and density, rendered into multi-view images conditioned on camera pose $c$, described as $X = G(w, c)$.

The NeRF-based inversion encoder $E$ (Li et al., 2023; Lan et al., 2023) learns the features of a large number of images with different poses to reconstruct 3D representation from a single image. The encoder maps an input image $X$ to the latent code $w$, which can be used to produce a novel view $X'$ of the input image :

$$
\begin{aligned}
w &= E(X), \\
X' &= G(E(X), c).
\end{aligned}
\tag{1}
$$

where $c$ is the camera pose.

Performing the diffusion process in the latent space combines the strength of other generators and accelerates training, especially in the 3D generation area. Here, there are several candidate latent spaces, such as $\mathcal{Z}$, $\mathcal{W}$, and $\mathcal{W}+$. The $\mathcal{W}$ space consists of $k$ repetitive 512-dimensional latent vectors fed into convolutional layers of different resolutions, but the $\mathcal{W}+$ space consists of $k$ distinct latent vectors. It's demonstrated that $k$ distinct latent vectors can increase the representation capacity of the generator than $k$ identical latent vectors (Shen et al., 2020; Li et al., 2023). Although EG3D is trained on the real-world face dataset FFHQ, our simple experiments in Appendix A.2.2 demonstrate that it is possible to generate some out-of-domain 3D faces, such as bronze statues, and cartoons, using fixed EG3D fed $\mathcal{W}+$ latent code. Moreover, $\mathcal{W}+$ space is considered more disentangled than $\mathcal{Z}$ and $\mathcal{W}$ spaces (Tov et al., 2021; Patashnik et al., 2021). Therefore, we choose $\mathcal{W}+$ space to perform the diffusion process in our work.

**Diffusion model architecture.** To obtain the latent code $w$ guided by human instructions, we employ a diffusion model to learn the correlation between the paired images and the instructions. Notably, transformers (Vaswani et al., 2017) show a promising ability to capture complex interactions and dependencies between various modalities (Bao et al., 2023). In this paper, our diffusion backbone is Diffusion Transformer (DiT) (Peebles & Xie, 2022). We have made the following modifications to DiT: (i) add an input header that enables paired latent codes diffusion, (ii) add a

cross-attention block after the self-attention block and introduce CLIP text embedding here, and (iii) add an identity embedding module that is plugged into the norm layer of transformer block using adaptive layer norm described in section 4.3.

Given an input image $X_o$ and a human instruction $T$, our goal can be formulated as learning the conditional distribution $p(w|X_o, T)$. The inversion encoder $E$ is applied to obtain the latent code $w_o = E(X_o) \in \mathbb{R}^{14*512}, w_e = E(X_e) \in \mathbb{R}^{14*512}$ for the original image $X_o$ and corresponding 2D edited image $X_e$ respectively. We add noise to the latent code $w_e$ with a fixed schedule, producing a noisy version $w_{et}$ at step $t, t \in T$. Our model is trained as a noise predictor $\epsilon_\theta$ to predict the added noise conditioned on image and text. The image conditioning $c_I$ consists of two parts: concatenation of $w_o$ and $w_e$, and identity modulation. The text conditioning $c_T$ is realized by adding a multi-head cross-attention block with the CLIP text embedding following the multi-head self-attention block. To adapt to the model, the latent code would be reshaped. Take 512*512 resolution for example, after padding 2 zero vectors for $w \in \mathbb{R}^{14*512}$, we reshape the latent code to the shape $512 * 4 * 4$. The conditional latent diffusion objective is:

$$\mathcal{L}_{diff} = \mathbb{E}_{w_e, c_I, c_T, \epsilon \sim \mathcal{N}(0,1), t}[\|\epsilon - \epsilon_\theta(w_{et}, t, c_I, c_T)\|_2^2], \tag{2}$$

where $c_I$ is the image conditioning, $c_T$ is the text conditioning.

## 4.2 Token Position Randomization

In our preliminary experiments, we have observed that when editing with multiple instructions, the more forward-positioned instructions are easier to show up in the edited image. We analyze this issue and attribute it to the training data being single instruction edited. In natural language processing, text conditioning requires that text be tokenized into a sequence of tokens, which is 77 in length in this paper. Only the first few sequence positions are usually non-zero when we train with single instruction data. It is intuitive to think that the cross-attention mechanism might pay more attention to the head of multiple instructions.

To achieve better editing results for the multiple instructions, we propose *token position randomization*, randomly setting the starting position of the text instruction tokens. This strategy makes the model more balanced in its attention to the components of multiple instructions. As a result, multi-semantic editing can be performed in a way that fulfills all the editing requirements while preserving identity well. In our paper, we randomly set the starting position in $[0, 30]$, with the last non-zero token position being less than 77. Ablation studies in Figure 5, and Table 3 show the effectiveness of the token position randomization strategy.

## 4.3 Identity Consistency Module

For precise face editing, preserving the input subject's identity is challenging, especially in 3D space. Latent code can be considered as compression of an image that loses some information, including identity. To tackle this, We impose an identity compensation module that directly modulates the extracted identity information into the diffusion process. A two-layer MLP network maps the identity features extracted from the original face into the same dimension as the diffusion timestep embedding. We regress dimensionwise scale and shift parameters $\gamma$ and $\beta$ from the sum of the embeddings of diffusion timestep $t$, and the embeddings of identity feature extracted from the portrait using a face identification model (Deng et al., 2021).

To improve 3D identity consistency further, we explicitly encourage face identity consistency at different poses by adding identity regularization loss between the 2D edited image and the rendered image with $yaw = 0$, and $pitch = 0$.

$$\mathcal{L}_{ID} = 1 - \langle F(X_e), F(G(\tilde{w}_{e0}, c_0)) \rangle, \tag{3}$$

where $F$ is the face identification model which extracts the feature of the face, $X_e$ is the 2D edited face, $\tilde{w}_{e0}$ is the one-step prediction of latent code, $c_0$ is the camera pose with yaw=0, pitch=0.

When the diffusion timestep is large, it will lead to a poor w for one-step prediction. Thus, this loss is calculated only for samples with timestep $t$ less than the timestep threshold $t_{th}$. The total loss is:

$$\mathcal{L} = \mathcal{L}_{diff} + \lambda_{id}\mathcal{L}_{ID}, \tag{4}$$

where $\lambda_{id}$ is the weight of $\mathcal{L}_{ID}$. $\lambda_{id}$ is set to 0.1 in our experiments.

Table 2: Quantitative evaluation for editing with single instruction on multi-view faces. ID score denotes the identity consistency between before and after editing. CLIP score measures how well the editing result matches the editing instructions. Attribute altering (AA) measures the change of the desired attribute. Attribute dependency (AD) measures the change in other attributes.

| Attribute | Instruction example | Method | ID↑ | CLIP↑ | AA↑ | AD↓ |
|-----------|--------------------|--------|-----|-------|-----|-----|
| Bangs | Let's add some bangs | Talk-To-Edit* | 0.41 | 0.07 | 0.97 | 0.56 |
| | | InstructPix2Pix | 0.40 | 0.07 | 0.80 | 0.64 |
| | | img2img | 0.40 | 0.10 | 0.99 | 0.61 |
| | | Ours | **0.56** | **0.13** | **1.05** | **0.53** |
| Eyeglasses | Make the person wearing glasses | Talk-To-Edit* | 0.46 | 0.04 | 0.69 | 0.67 |
| | | InstructPix2Pix | 0.51 | 0.17 | 3.27 | 0.65 |
| | | img2img | 0.42 | 0.17 | 3.33 | 0.79 |
| | | Ours | **0.59** | **0.20** | **3.37** | **0.64** |
| Smile | The person should smile more happily | Talk-To-Edit* | 0.43 | 0.10 | 0.54 | 0.62 |
| | | InstructPix2Pix | 0.48 | 0.16 | 1.46 | 0.66 |
| | | img2img | 0.46 | 0.15 | 1.47 | 0.76 |
| | | Ours | **0.60** | **0.18** | **1.50** | **0.61** |

*: Talk-To-Edit don't recognize some instructions. Only the results of recognized instructions are considered.

## 4.4 IMAGE AND TEXT CONDITIONING

Classifier-free guidance trains an unconditional denoising diffusion model together with the conditional model instead of training a separate classifier (Ho & Salimans, 2021). Liu et al. (2022) show that composable conditional diffusion models can generate images containing all concepts conditioned on a set of concepts by composing score estimates. Following (Ho & Salimans, 2021; Liu et al., 2022; Brooks et al., 2023), we train a single network to parameterize the image-text conditional, the only-image conditional, and the unconditional model. We train the unconditional model simply by setting $c_I = \emptyset, c_T = \emptyset$ with probability $p_1$, similarly, only setting $c_T = \emptyset$ with probability $p_2$ for the only-image-conditional model $\epsilon_\theta(w_{ot}, c_I, \emptyset)$. In our paper, we set $p_1 = 0.05, p_2 = 0.05$ as hyperparameters. In the inference phase, we add a little bit of noise to the latent of the input image (usually 15 steps) to obtain $w_{ot}$ and then use our model to perform conditional denoising. The model predicts three score estimates, the image-text conditional $\epsilon_\theta(w_{ot}, c_I, c_T)$, the only-image conditional $\epsilon_\theta(w_{ot}, c_I, \emptyset)$, and the unconditional $\epsilon_\theta(w_{ot}, \emptyset, \emptyset)$. $c_T = \emptyset$ indicates that the text takes an empty character. $c_I = \emptyset$ means that the concatenation $w_o$ takes zero and identity modulation takes zero. Image and text conditioning sampling can be performed as follows:

$$\begin{aligned}
\tilde{\epsilon}_\theta(w_{ot}, c_I, c_T) = &\epsilon_\theta(w_{ot}, \emptyset, \emptyset) \\
&+ s_I(\epsilon_\theta(w_{ot}, c_I, \emptyset) - \epsilon_\theta(w_{ot}, \emptyset, \emptyset)) \\
&+ s_T(\epsilon_\theta(w_{ot}, c_I, c_T) - \epsilon_\theta(w_{ot}, c_I, \emptyset))
\end{aligned} \tag{5}$$

where $s_I$ and $s_T$ are the guidance scales for alignment with the image and the text, respectively.

## 5 EXPERIMENTS

Given that the field of instructed 3D-aware face editing is under-explored, we have designed and developed a series of models that serve as our baselines. We compare our method with three baselines: Talk-To-Edit (Jiang et al., 2021) combined with PREIM3D (Li et al., 2023), InstructPix2Pix (Brooks et al., 2023) combined with PREIM3D, and a method similar to the *img2img* mode of Stable Diffusion (Rombach et al., 2022). More implementation details are provided in the Appendix A.1.

We use the official pre-trained models and code for Talk-To-Edit, InstructPix2Pix, and PREIM3D in the comparison experiments. The metrics are calculated on the first 300 images from CelebA-HQ. To evaluate the 3D capability, we uniformly rendered 4 views from yaw angles between $[-30°, 30°]$ and pitch angles between $[-20°, 20°]$ for an input image. We evaluate our method using ID, CLIP, AA, AD, $M_d$, and $S_d$, defined in the Appendix A.1.3.

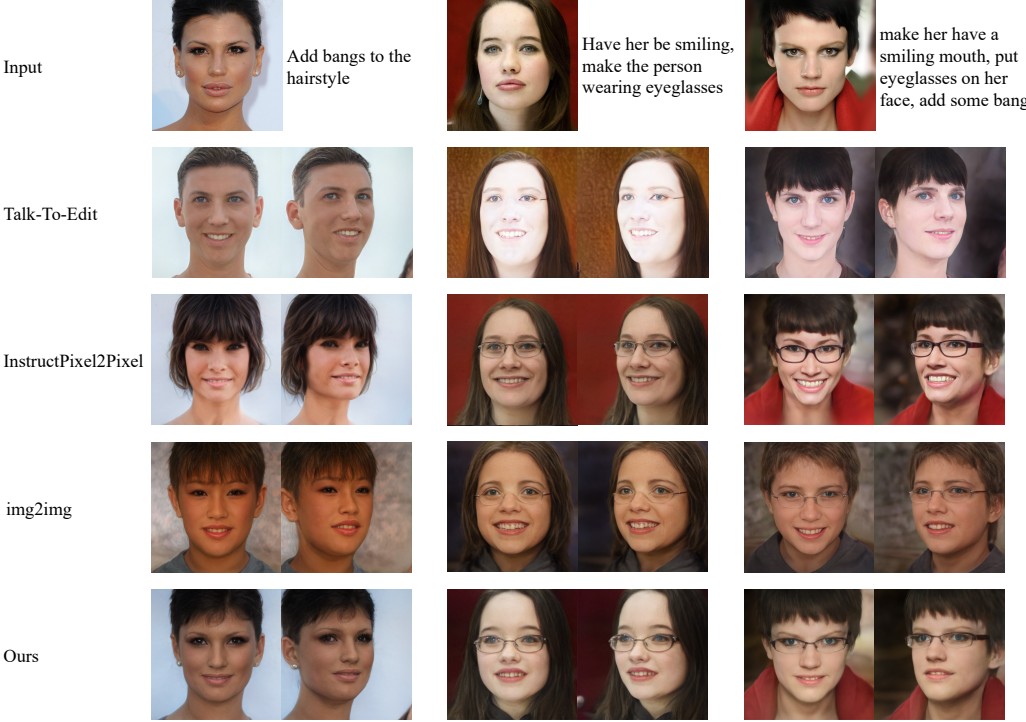

Figure 3: Qualitative comparison. Our method achieves the requirements of the text instruction while preserving identity consistency, especially multiple instruction editing.

Table 3: Quantitative evaluation for editing with multiple instructions on multi-view faces. $M_d$ and $S_d$ measure 3D consistency

| Method | ID↑ | CLIP↑ | $AA_{avg}$ ↑ | $AD_{avg}$ ↓ | $AA_{min}$ ↑ | $M_d$ ↓ | $S_d$ ↓ |
|---|---|---|---|---|---|---|---|
| Talk-To-Edit* | 0.44 | 0.05 | 0.24 | 0.58 | -0.12 | 0.125 | 0.042 |
| InstructPix2Pix | 0.46 | 0.19 | 1.48 | 0.71 | 0.31 | 0.109 | 0.039 |
| img2img | 0.37 | 0.17 | 1.39 | 0.88 | 0.12 | 0.119 | 0.039 |
| Ours(w/o TPR) | 0.50 | 0.18 | 1.50 | 0.73 | 0.08 | 0.114 | **0.038** |
| Ours | **0.55** | **0.20** | **1.53** | **0.69** | **0.52** | **0.105** | **0.038** |

*: Same as Table 2.

**Qualitative evaluation.** We present examples of the instructed editing results in Figure 3. Due to using a word bank, Talk-To-Edit does not recognize part of the instructions and cannot handle multiple instructions. Img2img does not disentangle the attributes and struggles in aligning text editing requirements with images, leading to changes in some other attributes. For example, the fourth row in Figure 3 shows the additional semantics of becoming younger. InstructPix2Pix usually leads to variations in hue and identity consistency, such as the skin tones in the third row of Figure 3. Our method achieves better text instruction correspondence, disentanglement, and 3D consistency than baselines. More editing results are provided in Appendix A.3

**Quantitative evaluation.** We chose three typical facial attributes, bangs, eyeglasses, and smile, to evaluate our method and baseline quantitatively. To be fair, our instruction test set consists of selected instructions from Talk-To-Edit, InstructPix2Pix, and InstructPix2NeRF, where each model contributes 15 instructions, such as *'Make her happy instead.'* and *'The eyeglasses could be more obvious.'* The metrics on the multiple instructions are measured with six combinations of the above three attributes in different orders. As shown in Table 2, 3, our method performs better than the baselines. Since img2img doesn't disentangle the editing requirements and is prone to cause changes on other attributes, it has a low ID score and a high AA score. Talk-To-Edit sometimes responds

only marginally to editing instructions, its CLIP and AA scores are significantly worse than other methods. InstructPix2Pix scores well, but still below our method. See Appendix A.2.3 for more evaluation of attribute editing. We conducted a user study and as shown in Table 6, our method outperforms the baselines.

Table 4: Effects of identity modulation and regularization loss. w/o $\mathcal{L}_{ID}$ and w/o ID cond represent the model without regularization loss and identity modulation, respectively.

| Config | $ID_{bang}$ | $ID_{eyeglasses}$ | $ID_{smile}$ | $ID_{multi}$ |
|---|---|---|---|---|
| w/o $\mathcal{L}_{ID}$ | 0.47 | 0.52 | 0.55 | 0.44 |
| w/o ID cond | 0.54 | 0.55 | 0.57 | 0.50 |
| Ours | **0.56** | **0.59** | **0.60** | **0.55** |

Put eyeglasses on him

Turn her into a Pixar character, put eyeglasses on her

instruction    input    w/o $L_{ID}$    w/o ID cond    Ours

Figure 4: Visual improvements of identity modulation and regularization loss.

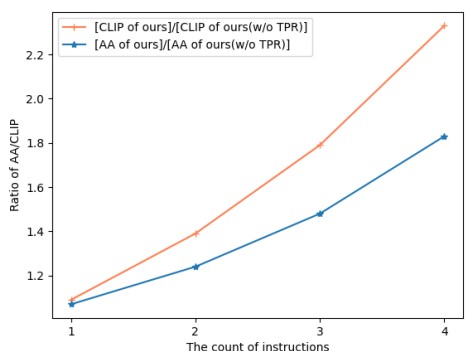

Figure 5: The improvement rate of AA and CLIP score for our model with a different number of editing instructions against the model without token position randomization training strategy.

**Ablation of token position randomization** To verify the effectiveness of token position randomization training, two models were trained, one using the token position randomization training strategy and the other not. Table 3 and Figure 8 show that the model with TPR performs better than the other model when editing with multiple instructions.

To further compare the performance of the two models for multiple instructions of different lengths, we performed multiple instructions editing of lengths 1-4, involving *bangs, eyeglasses, smile, age*. The average improvement in AA and CLIP scores measures the effect. As shown in Figure 5, the two models give comparable AA and CLIP scores when using single instruction editing. Still, as the instruction length increases, our model shows a better correspondence for text instruction.

**Ablation of identity consistency module** Injecting learned identity information into the diffusion process, the identity modulation compensates for losing information in latent space. The identity regularization loss explicitly guides the model to preserve the identity of the input subject. As shown in Table 4 and Figure 5, the model with the identity consistency module significantly improves the identity consistency scores and visuals.

## 6 CONCLUSIONS

In this paper, to solve the instructed 3D-aware face editing, we propose a novel conditional latent 3D diffusion model, enabling instructed 3D-aware precise portrait editing interactively. To support the editing of multiple instructions editing that was not available in previous methods, we propose the token position randomization training strategy. Besides, we propose an identity consistency module consisting of identity modulation and identity loss to improve 3D identity consistency. We expect our method to become a strong baseline for future works towards instructed 3D-aware face editing. Our method can be used for interactive 3D applications such as virtual reality and metaverse.

**Limitations.** One limitation of our work is that some semantically identical instructions may produce slight errors. For example, although "turn the hair color to pink" and "change her hair into pink" both want to change the hair color to pink, they will obtain different pink. Moreover, the same instruction will also have color differences for different people. We have some losses for details such as eye shape and eyelashes. These are issues that need to be addressed in this area in the future.

## ACKNOWLEDGEMENTS

This work was supported by NSFC Projects (Nos. 62350080, 62076147, U19A2081), BNRist (BNR2022RC01006), Tsinghua Institute for Guo Qiang, and the High Performance Computing Center, Tsinghua University. J.Z. was also supported by the New Cornerstone Science Foundation through the XPLORER PRIZE.

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

## A    APPENDIX

In the Appendix, we first provide implementation details, including the model parameters and training dataset. We follow with additional experiments and visual results. We highly recommend watching our video, which contains a live demonstration of interactive instructed editing.

### A.1    IMPLEMENTATION DETAILS

#### A.1.1    EXPERIMENT SETTING

We train our conditional diffusion model on the dataset we prepared from FFHQ (Karras et al., 2019) and use CelebA-HQ (Karras et al., 2018) for evaluation. In our experiments, we use pretrained EG3D model (Chan et al., 2022), pretrained PREIM3D model (Li et al., 2023), and pretrained 'ViT-H-14' CLIP model (Radford et al., 2021). The Diffusion Transformer is modified from the backbone of 'DiT B/1', adding an input header, a text condition module with the cross-attention mechanism, and an identity modulation module. The number of parameters in the model is 1.15 Billion. We set $t_{th} = 600$, $\lambda_{id} = 0.1$ and trained the model on a 4-card NVIDIA GeForce RTX 3090 for 6 days with a batch size of 20 on a single card.

#### A.1.2    TRAINING DATASET

The triplet data encourages the model to learn the correlation between the change from pairs of images and the instruction. Since the perceptual changes between the paired images are well-defined, we can easily generate corresponding instructions. Figure 6 shows the data preparation. For the paired images generated by e4e, we provide ChatGPT with some example data consisting of the paired attribute labels and a few corresponding handwritten transfer instructions. We guide ChatGPT by following these steps.

a. *You are now an excellent data generation assistant.*
b. *The rule for generating data is that I give you an input label and an output label, and you help me generate an instruction that represents the change from the input label to the output label.*
c. *These are some examples.*
*example 1*
    *input label: eyeglasses*
    *output label: without eyeglasses*
    *instruction: remove the eyeglasses.*
*example 2*
    *input label: no beard*
    *output label: beard man*
    *instruction: give him some beard*
*example 3*
    *input label: an old lady*
    *output label: a young girl*
    *instruction: make her look more youthful.*
*example 4*
    *input label: brown hair*
    *output label: blond hair*
    *instruction: turn the hair to blond*
d. *I will give you an input label and an output label to generate 10 institutions based on the rules and examples above.*
    *input label: a small nose*
    *output label: a big nose*

For the paired image generated by InstructPix2Pix, we guide ChatGPT like this, "*You are an excellent instruction generation assistant. I give you a face editing instruction, please generate 30 instructions that are semantically identical to this one.*".

Our dataset has about 40K paired images and the corresponding 3K instructions, with e4e generating 22K paired images and InstructPix2Pix generating 18K paired images. For the images generated by e4e, 500 images share about 30 instructions. For the images generated by InstructPix2Pix, 120 images share about 10 instructions. The resolution of the image is 512*512. All instructions are single editing requirements. We train our model with the dataset cropped as EG3D (Chan et al., 2022) and PREIM3D (Li et al., 2023). The dataset examples are provided in Figure 7.

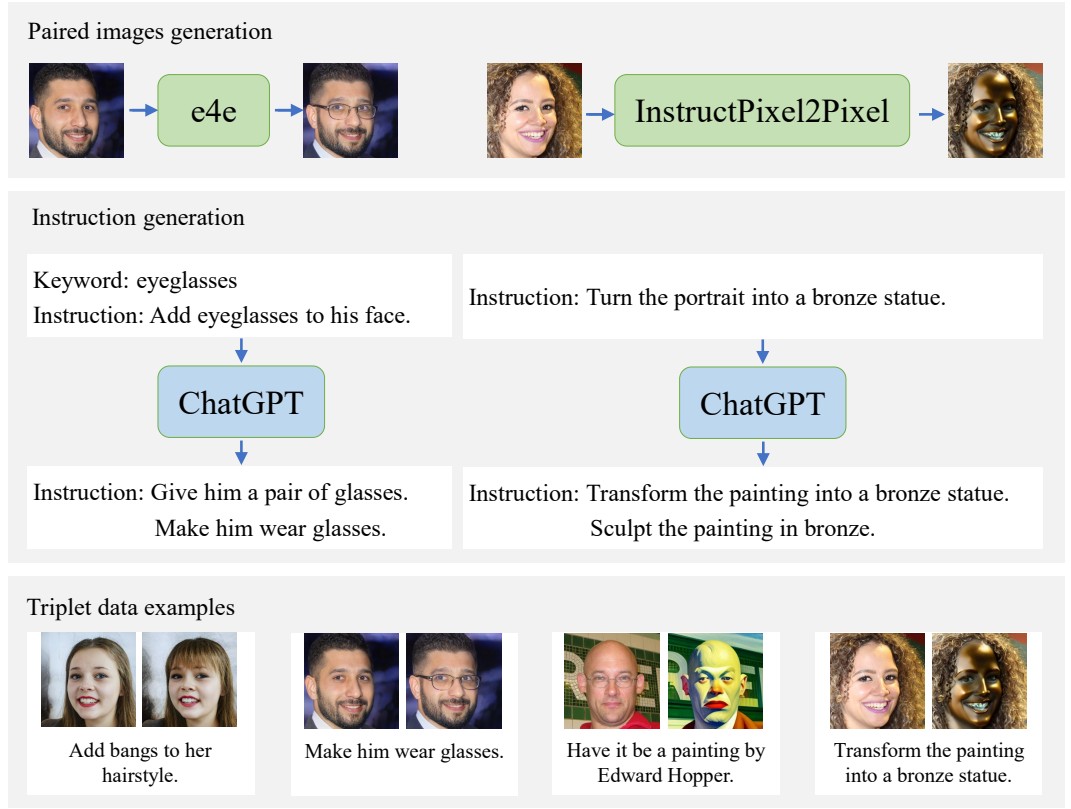

Figure 6: We use e4e and InstrucPixel2Pixel to generate paired images and use ChatGPT to produce the instructions.

### A.1.3 EVALUATION

**Test data.** The image test dataset is the first 300 images from CelebA-HQ (Karras et al., 2018). The instruction test set used in the comparison experiments consists of selected instructions from Talk-To-Edit, InstructPix2Pix, and InstructPix2NeRF, where each model contributes 15 instructions, such as '*Make her happy instead.*' and '*The eyeglasses could be more obvious.*'. The multiple instruction comprises 3 single instructions that are concated together. We show the test instructions in Table 7, 8, and 9.

**Baselines.** Talk-To-Edit and InstructPix2Pix are the most popular methods enabling instructed 2D editing. We perform 2D portrait editing guided by the instructions and then get the 3D-aware edited images using the state-of-the-art 3D inversion encoder PREIM3D. For another baseline img2img, we only trained a text-to-3D diffusion model using the instructions and edited images in the prepared dataset. Similar to the *img2img* mode of Stable Diffusion, we apply denoising sampling to the latent code of the input image with a small amount of noise added, conditional on the instruction.

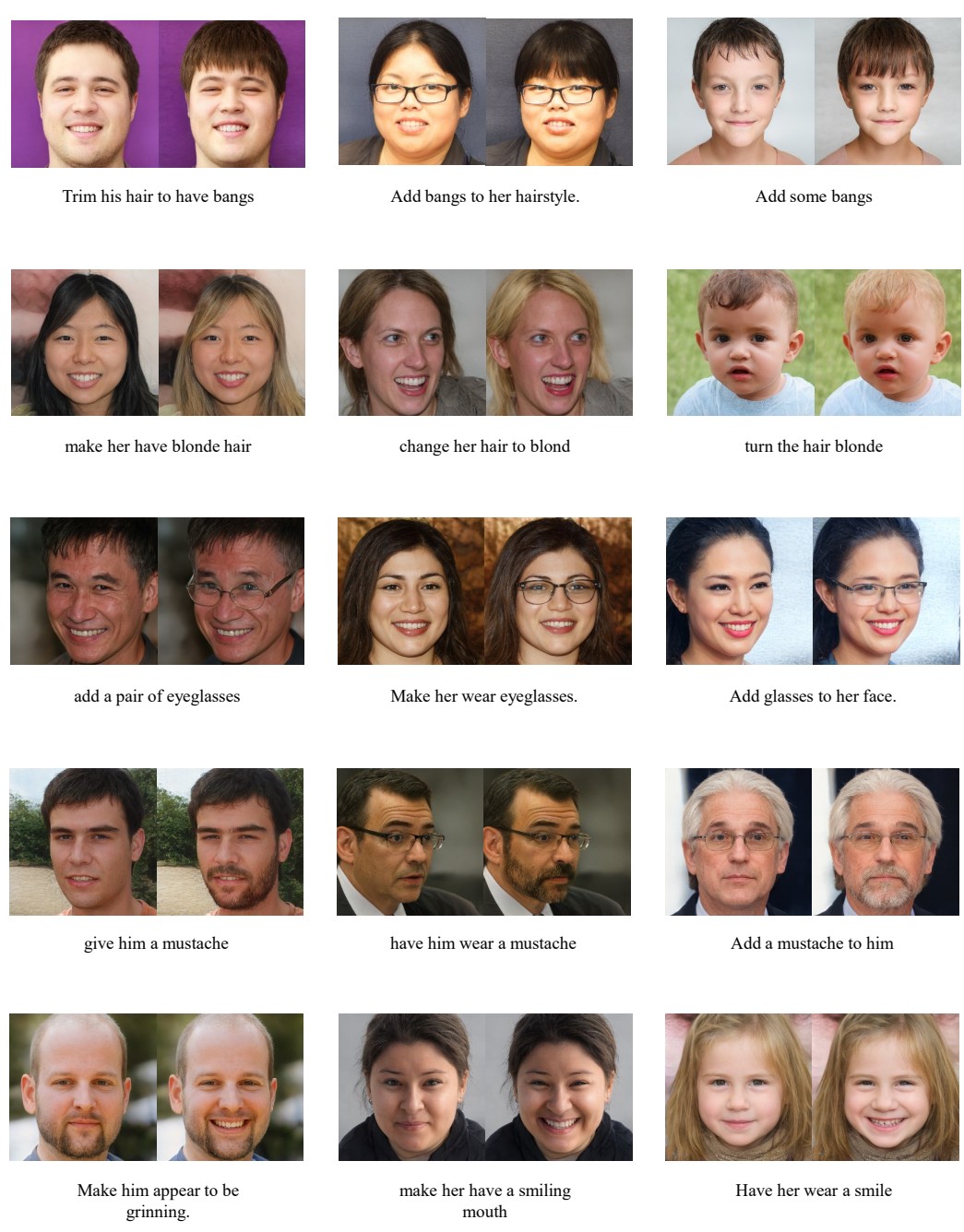

Figure 7: The dataset examples.

**Metrics.** The subject's multi-view identity consistency (ID) is measured by the average ArcFace feature similarity score (Deng et al., 2021) between the sampled images and the input image. We evaluate the precision of instructed editing with the directional CLIP similarity (Gal et al., 2022), which is calculated by the cosine similarity of the CLIP-space direction between the input image and multi-view edited images and the CLIP-space direction between the input prompt and edited prompt. Here, our input prompt is composed of attribute labels with a probability greater than 0.9 in an off-the-shelf multi-label classifier based on ResNet50 (Huang & Belongie, 2017), and the edited prompt is appended by the input prompt with the attribute label you want to edit. Following (Li et al., 2023; Wu et al., 2021), We use attribute altering (AA) to measure the change of the desired attribute

Style her hair with a pink wig, put eyeglasses on her.

Put eyeglasses on her, make her happy instead, turn the hair color to blue.

Put eyeglasses on her, create a smiling expression, and turn the painting into a bronze statue.

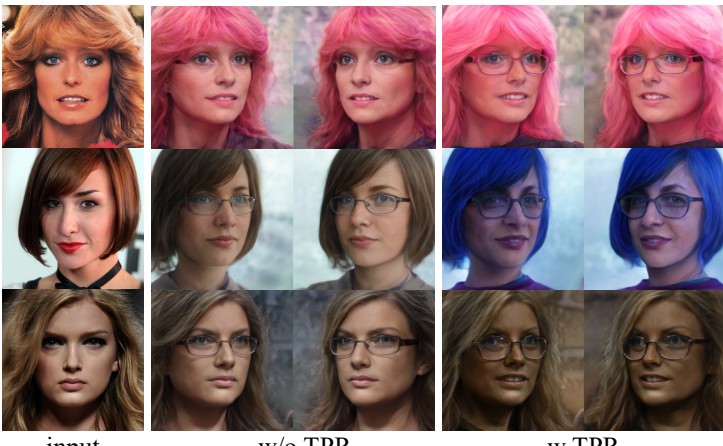

instruction      input      w/o TPR      w TPR

Figure 8: Editing results of multiple instructions. w/o TPR denotes the model without token position randomization training scheme.

and use attribute dependency (AD) to measure the change in other attributes when modifying a specific attribute. Attribute altering (AA) is the attribute logit change $\Delta l_t$ normalized by the standard deviation $\sigma(l)$ when detecting attribute $t$ by the classifier, and attribute dependency (AD) is the other attribute logit change. Following Abdal et al. (2023), we computed the mean differences ($M_d$) and standard deviation differences ($S_d$) metrics between the depth maps of the editing results and the depth map of the randomly sampled images in EG3D to measure 3D consistency.

## A.2 MORE EXPERIMENTS

### A.2.1 PROMPT-DRIVEN EDITING

Text-image pairing contributes significantly to text-to-image and image captioning. Focusing on adjectives and nouns in the text, prompt-driven editing methods perform better on descriptive prompts than instructed text. For example, StyleClip (Patashnik et al., 2021), the state-of-the-art text-guided 2D face editing method, usually misinterprets some verbs, as shown in Figure 9.

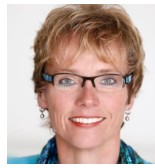 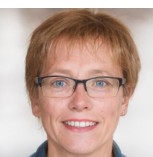 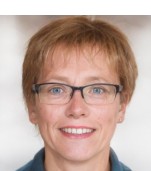 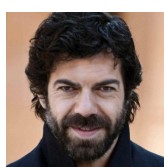 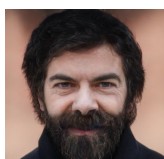 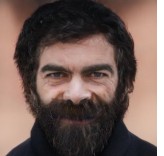

input     remove the eyeglasses     take off the eyeglasses     input     shave the beard     remove the beard

Figure 9: The instructed editing results of StyleClip.

### A.2.2 EXPLORING THE $W+$ SPACE OF EG3D

EG3D, the state-of-the-art NeRF-based generator in the face domain, is trained on a real-world face dataset. The original EG3D can generate high-resolution multi-view-consistent images and high-quality 3D geometry from the Gaussian noise. Although the original EG3D can only generate faces with a similar distribution to the FFHQ data, we get a wider range of reasonable face images by navigating in the $\mathcal{W}+$ space. The intermediate latent space of EG3D is a manifold of k identical 512-dimensional latent vectors. Using k distinct latent vectors rather than identical ones, we can greatly extend the generation capability of the fixed EG3D. Figure 17 illustrates our exploration of $\mathcal{W}+$ space in EG3D.

### A.2.3 MORE ATTRIBUTE EDITING

We evaluated the instructed editing performance of more attributes, as shown in Table 5. Our method outperforms InstructPix2Pix and img2img in AA, AD, and ID metrics.

Table 5: Quantitative evaluation for more instructed attribute editing.

| Instruction | five o'clock shadow | | | arched eyebrows | | | attractive | | | bags under eyes | | | bald | | |
|---|---|---|---|---|---|---|---|---|---|---|---|---|---|---|---|
| Method | IP2P | I2I | IP2N | IP2P | I2I | IP2N | IP2P | I2I | IP2N | IP2P | I2I | IP2N | IP2P | I2I | IP2N |
| ID↑ | 0.45 | 0.52 | **0.59** | 0.52 | 0.51 | **0.64** | 0.59 | 0.48 | **0.61** | 0.55 | 0.50 | **0.63** | 0.42 | 0.47 | **0.56** |
| AA↑ | 0.29 | 0.94 | **0.95** | -0.30 | 0.09 | **0.13** | -0.29 | -0.01 | **0.40** | 0.35 | **1.09** | 1.06 | 0.93 | 0.93 | **1.04** |
| AD↓ | 0.76 | 0.77 | **0.74** | 0.50 | 0.67 | **0.48** | 0.55 | 0.62 | **0.53** | 0.65 | 0.61 | **0.60** | 0.76 | 0.74 | **0.71** |
| Instruction | big lips | | | big nose | | | black hair | | | blond hair | | | brown hair | | |
| ID↑ | 0.53 | 0.50 | **0.59** | 0.42 | 0.49 | **0.59** | 0.48 | 0.54 | **0.63** | 0.54 | 0.47 | **0.60** | 0.51 | 0.52 | **0.62** |
| AA↑ | 0.37 | 0.37 | **0.44** | 0.42 | 0.65 | **0.97** | 1.05 | **1.12** | 1.09 | 0.96 | 0.87 | **0.98** | **0.36** | 0.14 | 0.28 |
| AD↓ | 0.53 | 0.52 | **0.49** | 0.75 | 0.66 | **0.65** | 0.59 | 0.60 | **0.52** | 0.58 | 0.64 | **0.55** | 0.58 | 0.61 | **0.56** |
| Instruction | bushy eyebrows | | | chubby | | | double chin | | | young | | | goatee | | |
| ID↑ | 0.56 | 0.53 | **0.64** | 0.48 | 0.50 | **0.64** | 0.48 | 0.51 | **0.64** | 0.43 | 0.47 | **0.50** | 0.57 | 0.47 | **0.61** |
| AA↑ | 0.89 | **1.02** | 1.00 | 0.98 | **1.05** | **1.05** | 0.94 | 0.96 | **1.00** | 0.21 | 0.01 | **0.77** | 1.25 | 1.30 | **1.40** |
| AD↓ | 0.75 | 0.62 | **0.51** | 0.59 | 0.65 | **0.54** | 0.76 | 0.76 | **0.63** | 0.68 | 0.65 | **0.63** | 0.80 | 0.85 | **0.78** |
| Instruction | gray hair | | | heavy makeup | | | high cheekbones | | | male | | | mouth open | | |
| ID↑ | 0.52 | 0.54 | **0.64** | 0.54 | 0.51 | **0.63** | 0.55 | 0.51 | **0.64** | 0.38 | 0.48 | **0.62** | 0.45 | 0.50 | **0.65** |
| AA↑ | **1.11** | 1.06 | 0.99 | 0.11 | 0.03 | **0.53** | 0.29 | 0.96 | **1.07** | 1.03 | **1.16** | 1.03 | 0.84 | 1.12 | **1.21** |
| AD↓ | 0.59 | 0.63 | **0.57** | 0.52 | 0.55 | **0.50** | 0.57 | 0.62 | **0.54** | 0.80 | 0.78 | **0.68** | 0.71 | 0.57 | **0.50** |
| Instruction | mustache | | | narrow eye | | | no beard | | | pale skin | | | pointy nose | | |
| ID↑ | 0.53 | 0.51 | **0.60** | 0.46 | 0.50 | **0.65** | 0.45 | 0.50 | **0.62** | 0.50 | 0.52 | **0.66** | 0.52 | 0.48 | **0.59** |
| AA↑ | **0.55** | 0.50 | 0.47 | 0.15 | 1.23 | **1.46** | -0.39 | -0.09 | **0.27** | 1.02 | 1.03 | **1.11** | -0.33 | -0.16 | **0.57** |
| AD↓ | **0.64** | 0.72 | 0.68 | 0.55 | 0.56 | **0.50** | 0.72 | 0.70 | **0.49** | 0.51 | 0.53 | **0.47** | **0.56** | 0.60 | 0.60 |
| Instruction | rosy cheeks | | | sideburns | | | lipstick | | | straight hair | | | wavy hair | | |
| ID↑ | 0.51 | 0.52 | **0.60** | 0.34 | 0.52 | **0.62** | 0.55 | 0.51 | **0.62** | 0.44 | 0.50 | **0.62** | 0.33 | 0.47 | **0.59** |
| AA↑ | -0.43 | 0.48 | **0.52** | 1.02 | 1.25 | **1.29** | 0.15 | 0.07 | **0.41** | **0.57** | 0.27 | 0.45 | 0.72 | 0.89 | **0.99** |
| AD↓ | **0.49** | 0.55 | 0.51 | 0.89 | 0.82 | **0.75** | 0.51 | 0.53 | **0.50** | 0.57 | 0.62 | **0.53** | 0.84 | 0.54 | **0.49** |

*note: IP2P, I2I, and IP2N denote InstructPix2Pix, img2img, and InstructPix2NeRF, respectively.*

### A.2.4 USER STUDY

To perceptually evaluate the instructed 3D-aware editing performance, we conduct a user study in Table 6. We collected 1,440 votes from 30 volunteers, who evaluated the text instruction correspondence and multi-view identity consistency of editing results. Each volunteer is given a source image, our editing result, and baseline editing, and asked to choose the better one, as shown in Figure 18 The user study shows our method outperforms the baselines.

Table 6: The result of our user study. The value represents the rate of Ours > others. Multiple instructions indicate editing with the combinations of the above three attributes.

| Method | bang | eyeglasses | smile | multiple instructions |
|---|---|---|---|---|
| Talk-to-Edit | 0.742 | 0.958 | 0.817 | 0.875 |
| InstructPix2Pix | 0.833 | 0.667 | 0.725 | 0.683 |
| img2img | 0.733 | 0.758 | 0.750 | 0.783 |

### A.2.5 W+ OPTIMIZATION ABLATION

In 2D image editing, although optimization-based inversion may be more time-consuming than encoder-based inversion, it produces better identity consistency. We replace the inversion encoder with latent optimization in our pipeline to see if it improves identity consistency. Since each optimization takes several minutes, we cannot perform optimizations during training, but only at inference time.

In our experiments, we considered two configurations: Direct $\mathcal{W}+$ optimization and PTI Roich et al. (2022) optimization. Direct $\mathcal{W}+$ optimization involves optimizing the $\mathcal{W}+$ vector while keeping the generator fixed. PTI (Pivotal Tuning Inversion) technique fine-tunes the generator based on the initial value provided by direct optimization. We conducted 500 steps of optimization on the $\mathcal{W}+$ vector, and PTI added 100 steps of fine-tuning the generator.

The results of these experiments are presented in Figure 10, where we compare the outcomes of direct $\mathcal{W}+$ optimization, PTI, and the encoder-based method. The results show that directly replacing the encoder with an optimization method during inference will lead to a severe decrease in both editing effect and identity consistency. We attribute this issue to the deviation between the model and data distribution. The model learns a conditional distribution within the encoder's inversion space during training. When the encoder is replaced by an optimization method during inference, the data distribution used for inference mismatches the learned model distribution. This mismatch results in greater identity drift and undesirable editing outcomes.

While conducting $\mathcal{W}+$ optimization during training (much larger compute) could potentially address the distribution deviation problem, it may introduce artifacts in novel views, as pointed out by PREIM3D. This is due to optimization being performed on a single image during training. In summary, while direct optimization of the $\mathcal{W}+$ vector is an interesting concept, our experiments suggest that it may not necessarily lead to improved identity preservation and editing results compared to the encoder-based approach.

### A.2.6 EFFECTS OF THE BACKGROUND

To verify the effect of background on the editing results, we edited images of the same person in different scenes. We show the results in Figure 11, where the first two rows are the same subject, the middle two rows are the same subject and the last two rows are the same subject. The results show that the background has no obvious impact on the editing results. However, note that when editing colors, particularly when the color being edited is close to the background color, there can be some blending between the foreground and background elements.

### A.2.7 THE TRIPLET DATA MECHANISM ABLATION

The triplet data mechanism plays a crucial role in achieving accurate and disentangled image editing. To provide a more thorough understanding of its importance, we conducted an ablation study. Our comparison involves the img2img model, which can be considered as using a text-image pairing data mechanism, in contrast to our method which utilizes the triplet data mechanism. Unlike InstructPix2NeRF, img2img has no paired images, but the rest of the network structure is the same.

The results of the ablation study, as shown in 3 and Table 2, 3, and 5, demonstrate that the triplet data mechanism significantly contributes to the quality of editing in terms of identity preservation (ID) and attribute dependency (AD) when attribute altering (AA) is close to equal.

Our method consistently outperforms img2img in preserving identity across various attributes, as indicated by the higher ID scores. Moreover, the triplet data mechanism helps reduce attribute dependency, ensuring that changes to one attribute do not excessively affect others. These results highlight that the triplet data mechanism encourages the model to learn the correlations between changes in pairs of images and the corresponding instructions, leading to more precise and disentangled editing. In conclusion, the triplet data mechanism is essential for achieving high-quality image editing results.

### A.3 MORE VISUAL RESULTS

We provide a large number of instructed editing results produced by InstructPix2NeRF in Figure 12, 13, 14, 15, and 16

**Diversity and generalization.** We realize the importance of diversity in the evaluation of image editing methods and strive to provide a comprehensive evaluation. As described in Appendix A.1.1 Experimental Settings, our model is trained on the FFHQ dataset, which consists of 70,000 high-quality faces featuring vast diversity in terms of age, ethnicity, and background, while also exhibiting comprehensive representation of accessories such as glasses, sunglasses, and hats. This diverse

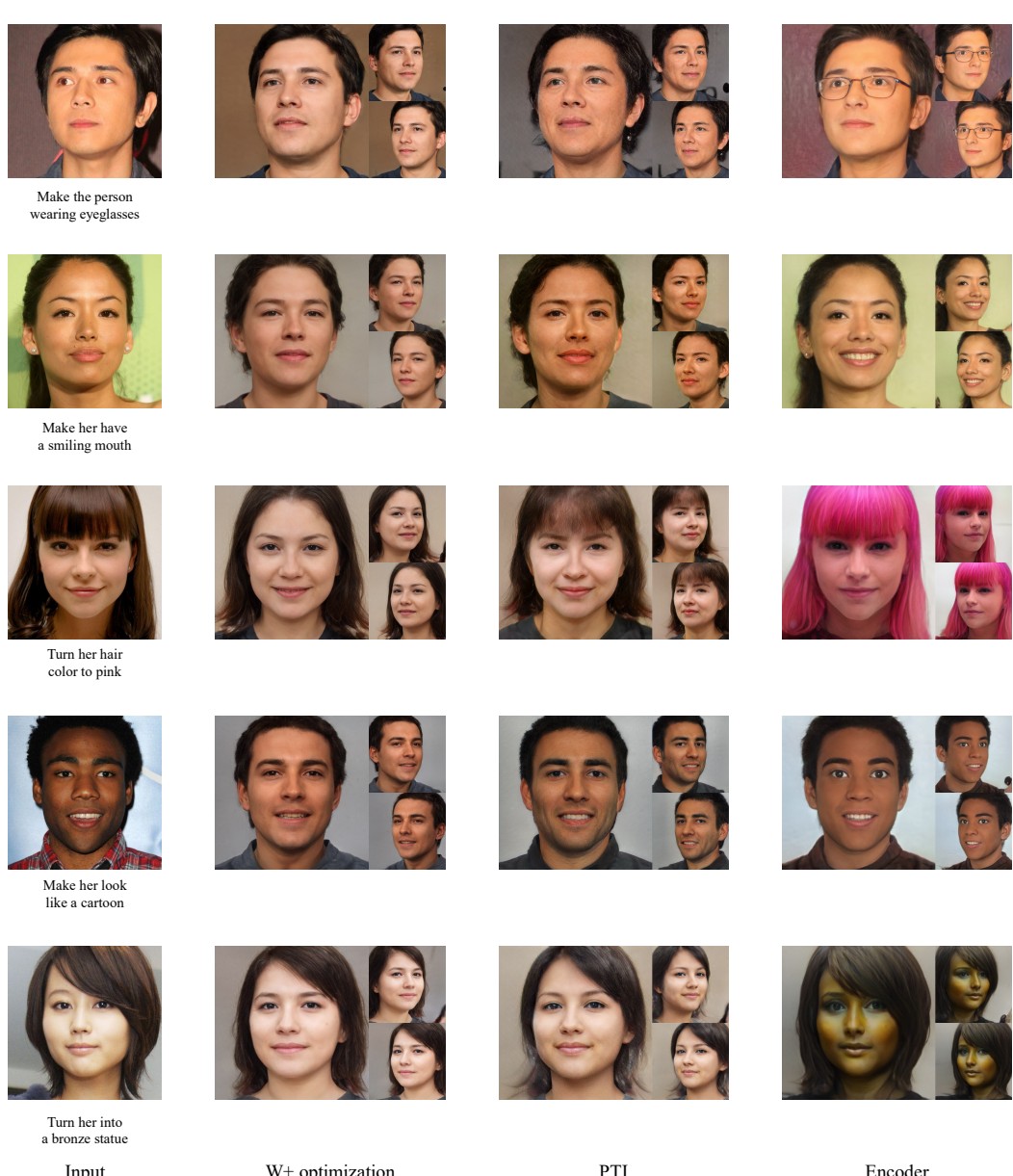

Figure 10: Replacing the inversion encoder with the optimization method.

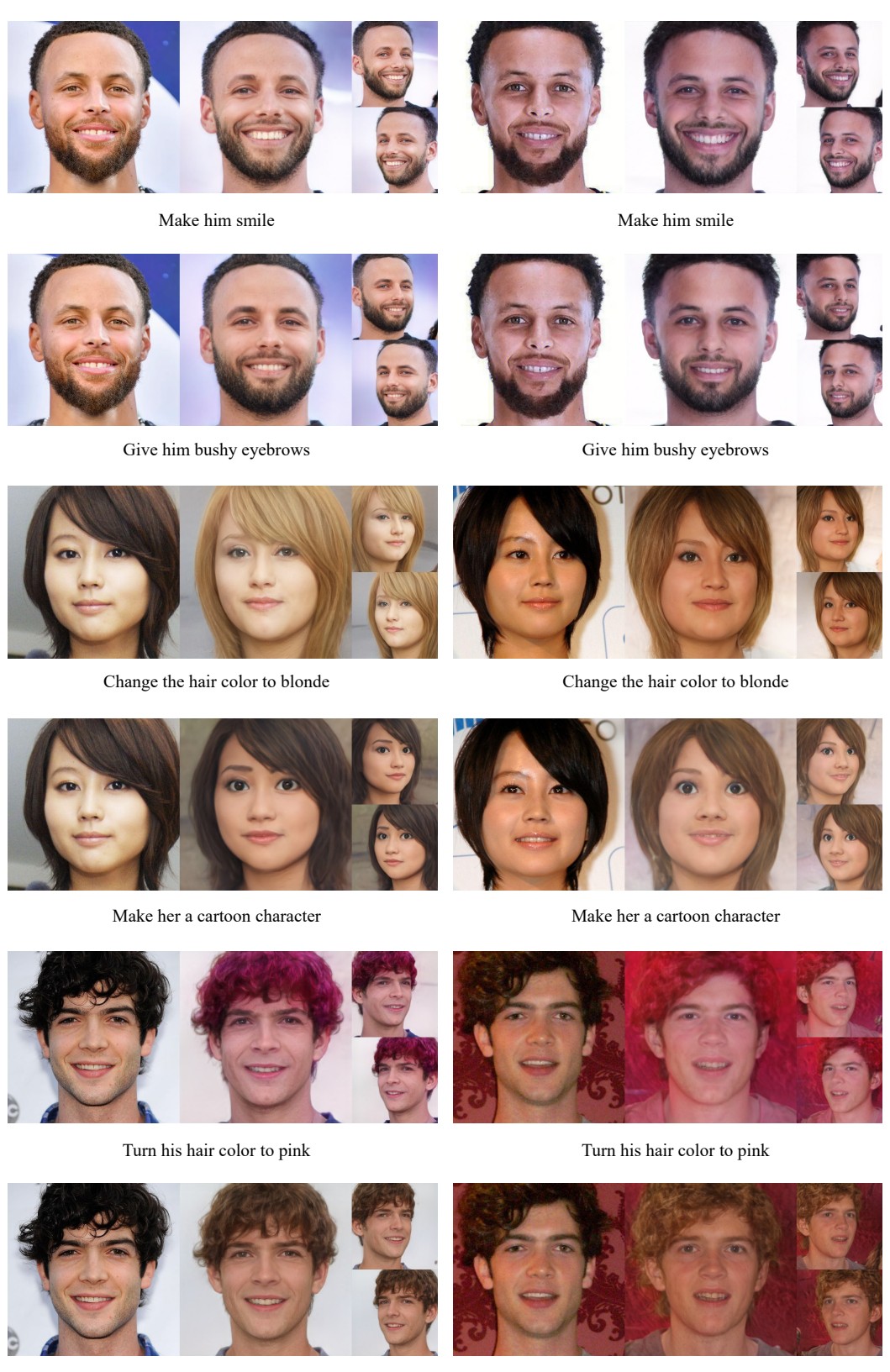

Figure 11: The effects of background.

Table 7: The test human instructions on bangs.

| Source | Instruction |
|---|---|
| InstructPix2Pix | Add a fringe to the wig.
Make the hairstyles have a fringe.
Make the bangs fringes.
Add a fringe.
Flapper hairstyles with a fringe. |
| Talk-To-Edit | Let's try long bangs.
What about adding longer bangs?
Emm, I feel the bangs can be longer.
How about trying adding longer bangs?
The bangs should be much longer. |
| InstructPix2NeRF | Style his hair with bangs.
Move the bangs to the front.
Let's add some bangs.
Give the girl bangs.
Add bangs to the hairstyle. |

Table 8: The test human instructions on eyeglasses.

| Source | Instruction |
|---|---|
| InstructPix2Pix | with glasses
Make the person wearing glasses
Make him wear glasses.
Have the faces be wearing glasses.
The women are wearing eyeglasses |
| Talk-To-Edit | Make the eyeglasses more obvious.
How about trying thick frame eyeglasses.
It should be eyeglasses with thin frame.
The eyeglasses could be more obvious.
Try eyeglasses. |
| InstructPix2NeRF | Make her wear glasses.
Give her a pair of eyeglasses.
Apply glasses to her face.
Place eyeglasses on her nose.
Add a pair of eyeglasses |

training data ensures that our model can generalize to various races and attributes. As shown in Figures 14 and 15, our models are capable of handling different races, hairstyles, ages, and other attributes. In Figure 16, we demonstrate this capability using the results of editing scenarios featuring characters from this year's movie "Mission: Impossible – Dead Reckoning Part One."

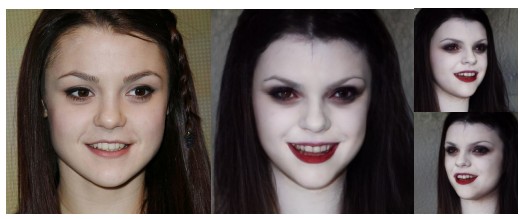
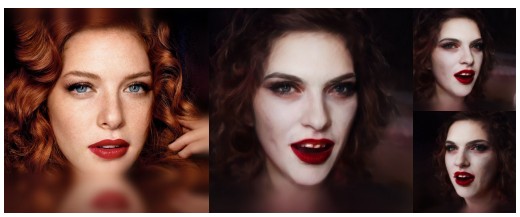

Turn her into a vampire                    Turn her into a vampire

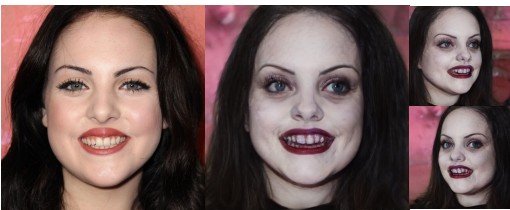
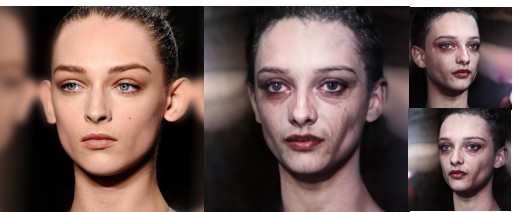

Give it a zombie makeover                  Give it a zombie makeover

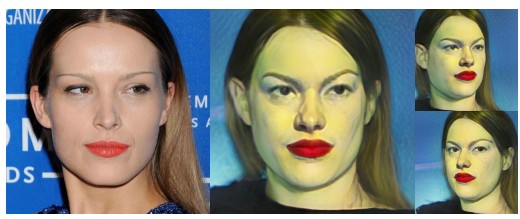
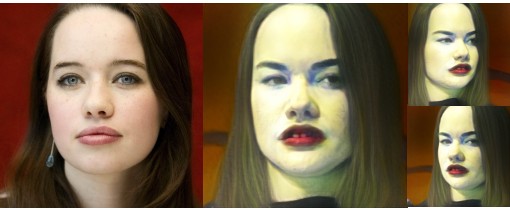

Make it look like a sketch by Edward       Make it look like a sketch by Edward
Hopper                                     Hopper

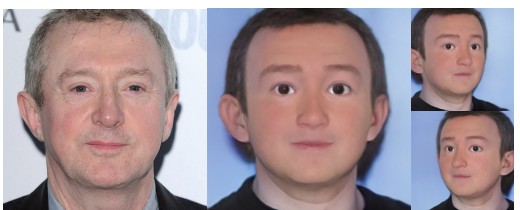
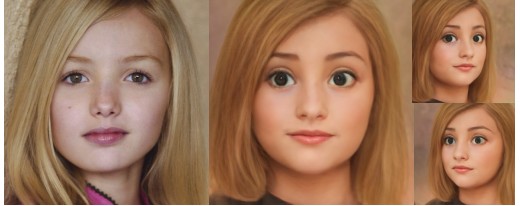

Make him a cartoon character               Make her a cartoon character

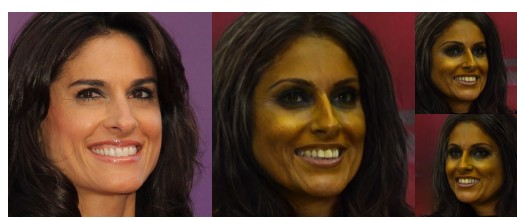
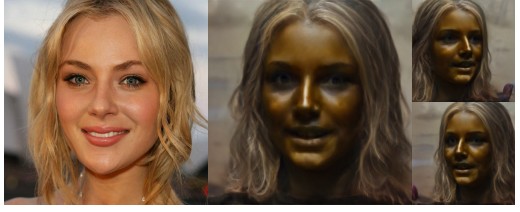

Turn the painting into a bronze statue     Turn the painting into a bronze statue

Figure 12: More Visual Results.

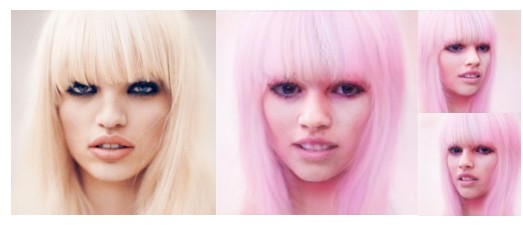 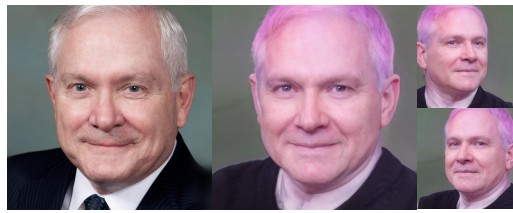

Turn the hair color to pink

Turn the hair color to pink

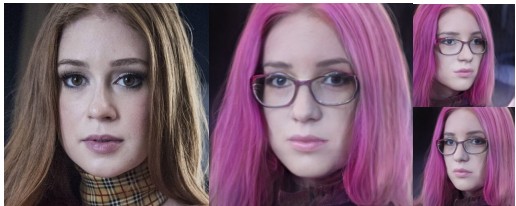 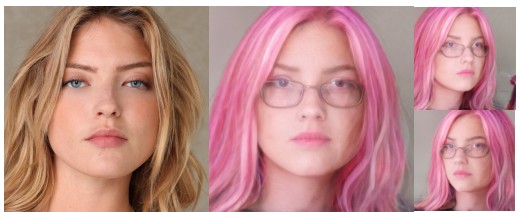

Turn the hair color to pink, put eyeglasses
on her

Turn the hair color to pink, put eyeglasses
on her

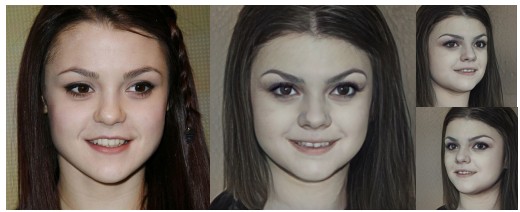 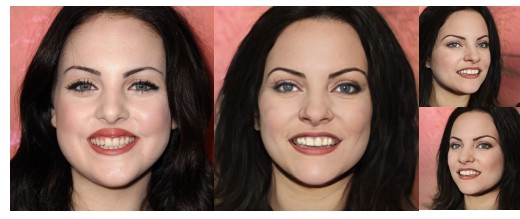

Give the portrait a comic book look

Give the portrait a comic book look

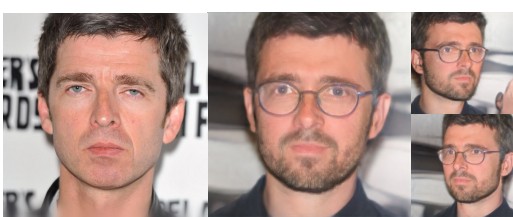 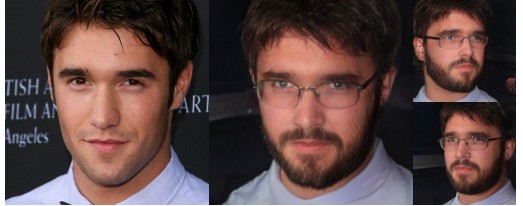

Put eyeglasses on his face, give him a
goatee

Put eyeglasses on his face, give him a
goatee

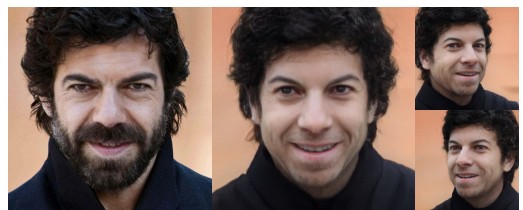 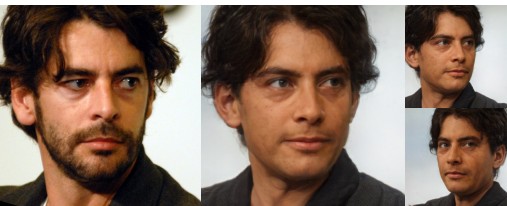

Remove the beard

Remove the beard

Figure 13: More Visual Results.

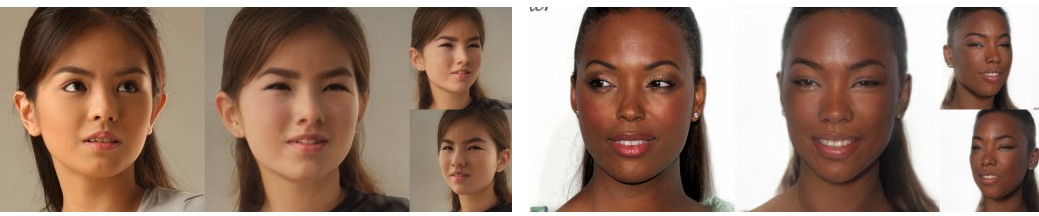

Make her eyes appear narrower          Make her eyes appear narrower

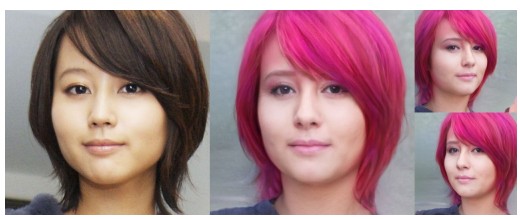 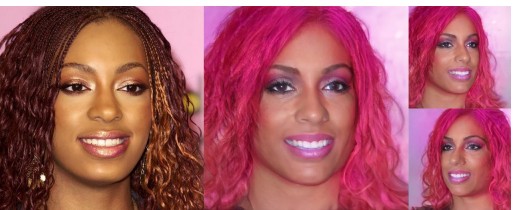

Style her hair with a pink wig          Style her hair with a pink wig

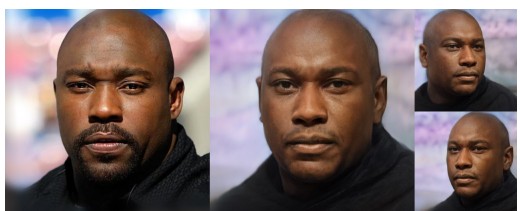 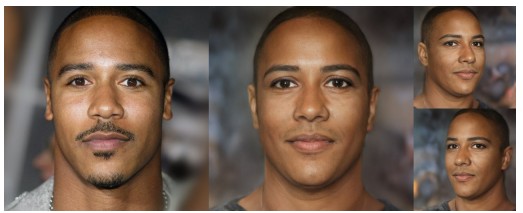

Shave the beard off          Shave the beard off

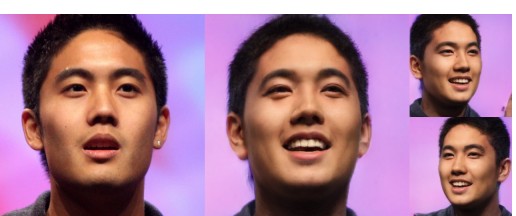 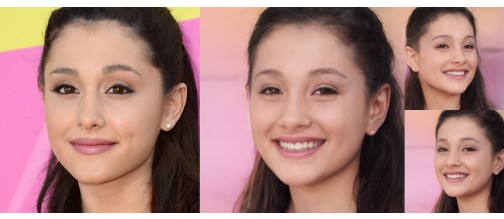

Have his wear a smile          Add a smile to her face

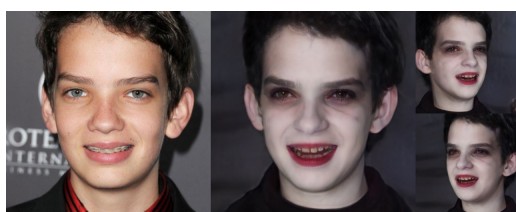 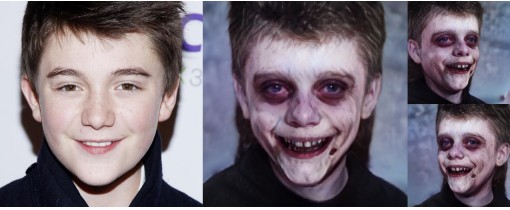

Turn him into a vampire          Give it a zombie makeover

Figure 14: More Visual Results.

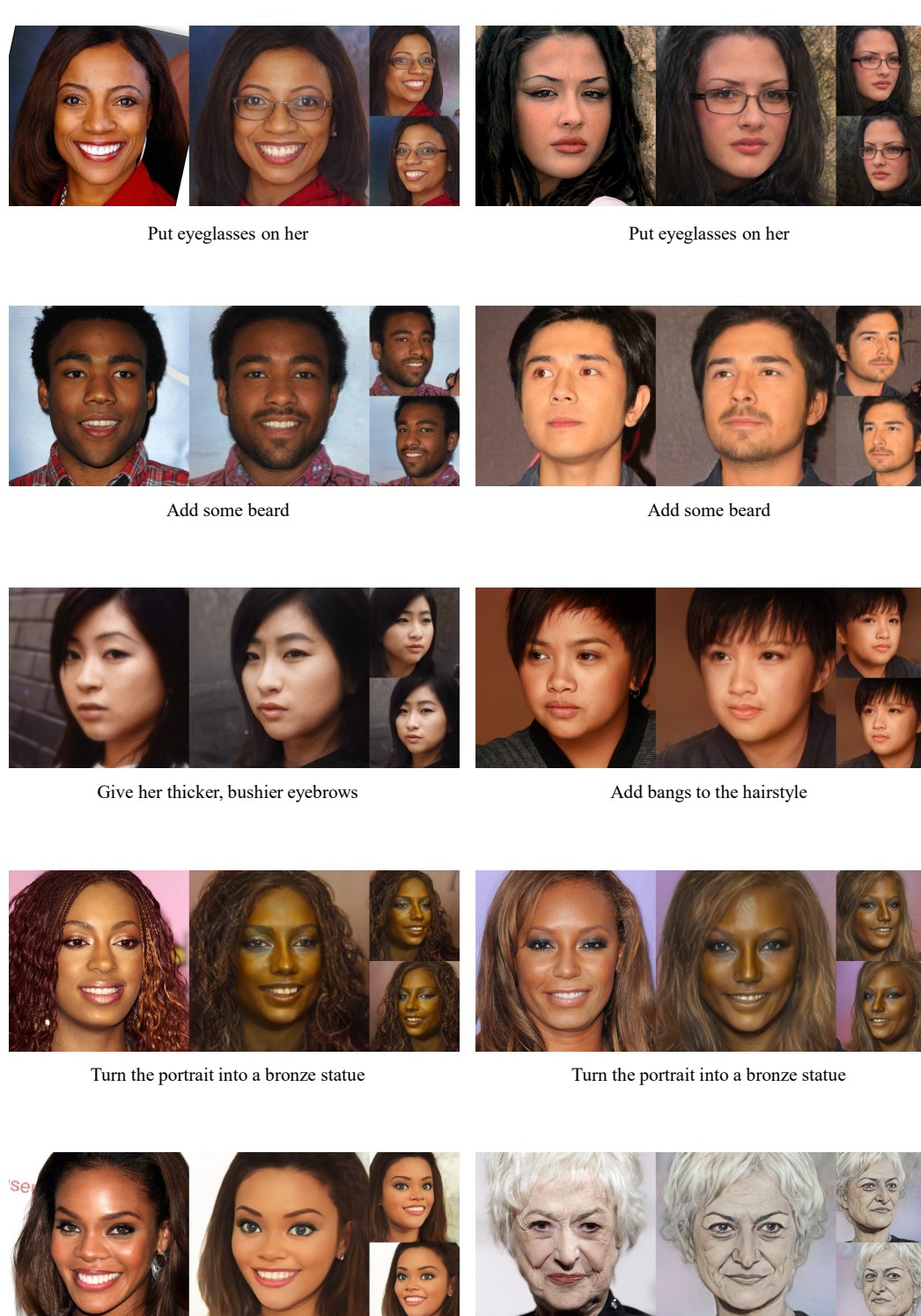

Figure 15: More Visual Results.

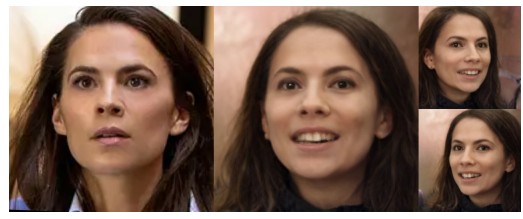 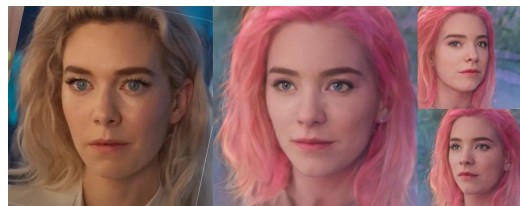

Give her a happy expression                    Turn her hair to pink

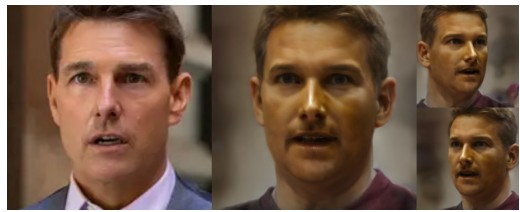 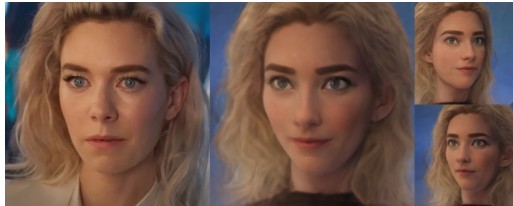

Turn the portrait into a bronze statue         Make her a cartoon character

Figure 16: Characters in Mission: Impossible – Dead Reckoning Part One.

Table 9: The test human instructions on smile.

| Source | Instruction |
|---|---|
| InstructPix2Pix | Have her be smiling
The faces should be smiling
Have an open-mouthed shark smiling
Add a "smiling" emoji
Make her happy instead |
| Talk-To-Edit | The person should smile more happily.
The person can smile to some degree more happily.
I kind of want the face to be smiling with the mouth wide open.
I would like the face to be smiling with teeth visible.
I would like to change the pokerface face to a smiling face. |
| InstructPix2NeRF | Alter her expression to appear cheerful and merry.
Give her a happy expression with a smile and bright eyes.
Adjust her features to convey a sense of happiness and positivity.
Make her look cheerful and full of good cheer.
Make the person have a smiling face |

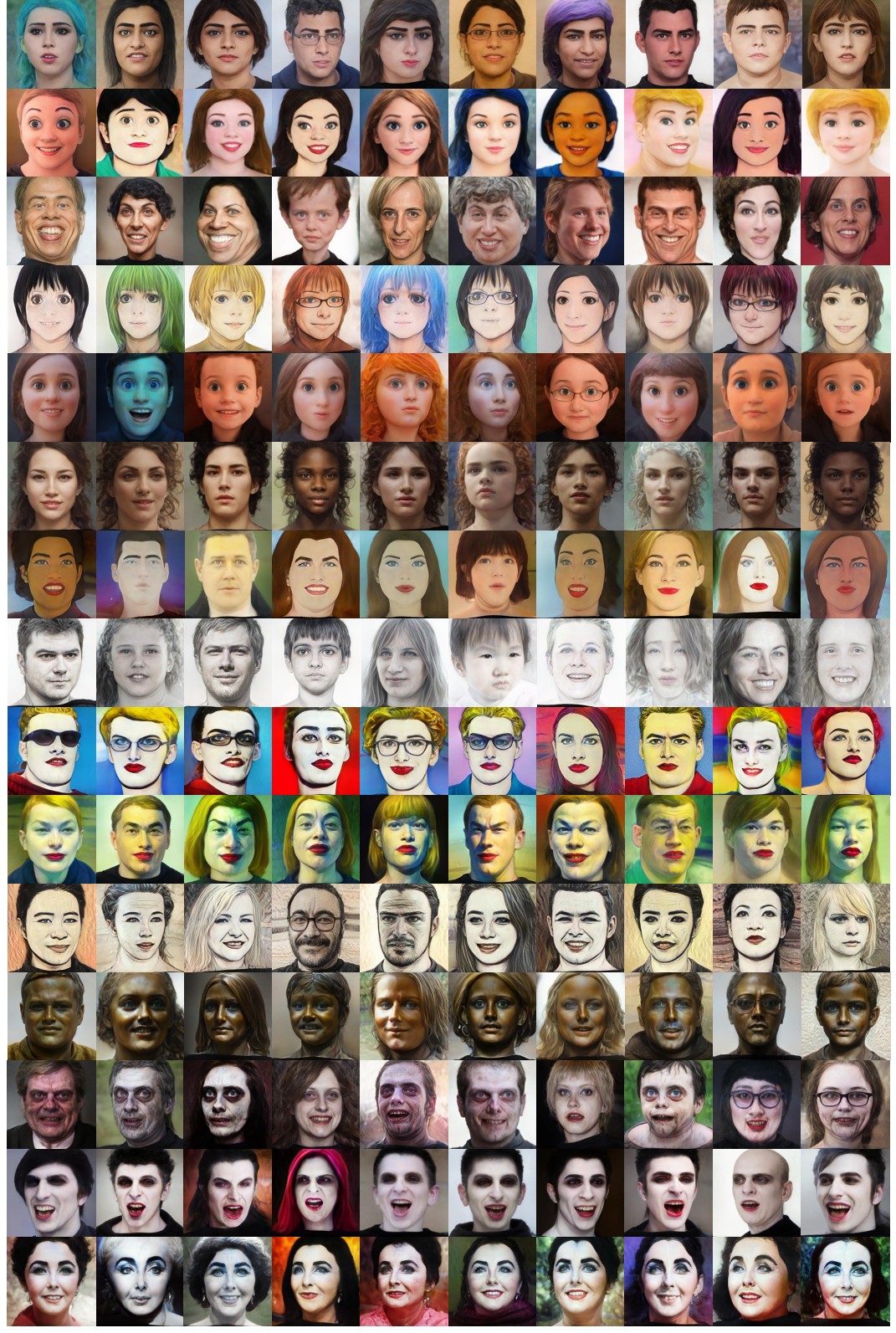

Figure 17: Our exploration of $\mathcal{W}+$ space.

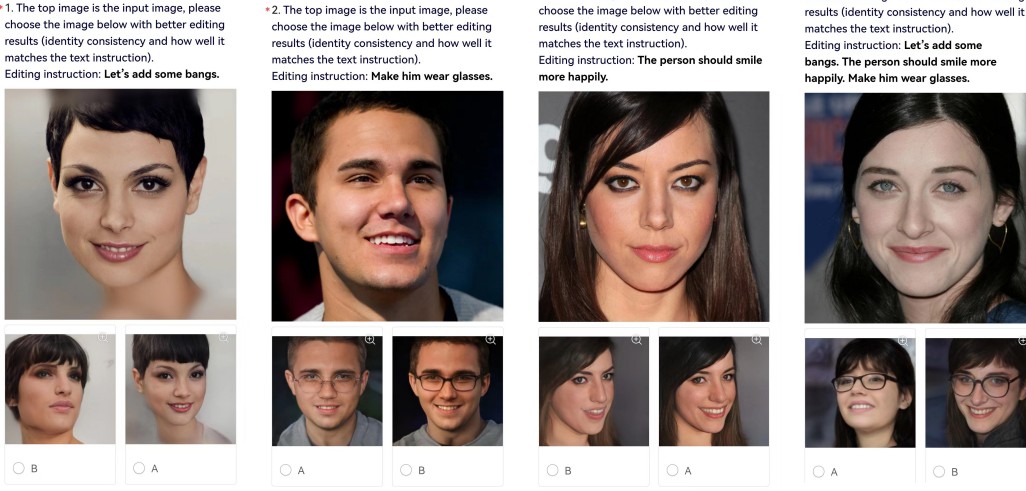

Figure 18: Examples from user study, the first choice (A: Ours, B: InstructPix2Pix), the second choice (A: img2img, B: Ours), the third choice (A: ours, B: Talk-To-Edit), the fourth choice (A: InstructPix2Pix, B: Ours)

