# OpenReview forum: "InstructPix2NeRF: Instructed 3D Portrait Editing from a Single Image"
_ICLR.cc/2024/Conference — ICLR 2024 poster_

### Official Review · Reviewer_1rfe · 2023-10-28

**Soundness:** 2 fair
**Presentation:** 3 good
**Contribution:** 2 fair
**Rating:** 6
**Confidence:** 3

**Summary:**

This paper proposes a method that generates a 3D-edited NeRF from a single portrait image. The style is defined by an instructing prompt. Two streams of inputs of a real face and a 2D-edited face are passed through an encoder to generate identity conditions. The identity condition, together with text condition, is sent to a diffusion model to generate tri-plane features for NeRF rendering. Experimental results show that the proposed method outperforms compared baseline approaches under the authors' settings.

**Strengths:**

- It is the first (to my knowledge) paper that allows "instructed" 3D portrait editing from single images.

- The experiments show that the proposed method outperforms compared baselines under the authors' settings.

**Weaknesses:**

- The results shown in the paper lack race diversity. There are almost no Asian or black people. I'm worried whether the proposed method does not perform well on those cases.

- The identity may change after applying the proposed method. For example, in Fig. 1 first example, the eye shape changed after the beard was removed. In Fig. 3 middle example, the girl seems to look more Asian and the nose shape changed after editing. These are not analyzed in the limitation section.

- The proposed method adopts two streams of inputs (real and edited images). However, the ablation study does not show the necessity of
 them. Will only one stream work?

**Questions:**

I would like to see the authors address my concerns mentioned in the weakness section.

---

> ### Author Response · Authors · 2023-11-19
> **Official Response to Reviewer 1rfe (1/2)**
>
> We greatly appreciate your recognition of the significance and novelty of our work. Below, we address your questions and concerns to provide a more comprehensive understanding of our approach.
>
> ### Summary of your key concerns and our improvements:
> 1. You suggest we provide examples of different races (W1)
> - **Response:** We provided additional examples featuring Asians and black people and discussed the diversity and generalization of our method in Appendix A.3.
> 2. You have some concerns about the preservation of identity. (W2)
> - **Response:** We have added more examples and comparative experiments and show the significant improvement of our method over previous methods through quantitative metrics and user study.
> 3. You suggest we discuss the necessity of the triplet data mechanism. (W3)
> - **Response:** We conducted an ablation study in Appendix A.2.7 and showed that the triplet data mechanism is essential for achieving high-quality image editing results.
>
> ### Detailed responses to your concerns item-by-item.
>
> > *W1: The results shown in the paper lack race diversity.*
>
> **A**:
> Thank you for raising this important point. We acknowledge the importance of diversity in the evaluation of image editing methods and strive to provide a well-rounded assessment. **Our models are indeed capable of handling different races, hairstyles, ages, and other attributes.**
>
> As described in Appendix A.1.1 Experimental Settings, our model is trained on the FFHQ dataset, which includes a diverse set of faces, and evaluated on the CelebA-HQ dataset. This diverse training data ensures that our model can generalize to various races and attributes.
>
> To address your concern, we have included editing results for both Asian and black people in Figures 14 and 15. These additional examples show the diversity of our method in handling different races.
>
> > *W2: The identity may change after applying the proposed method.*
>
> **A**: Thank you for your careful observation. Identity preservation and attribute disentanglement have posed challenges for face editing. 3D-aware editing becomes more difficult than 2D editing in this regard since only one input image is visible.
>
> The ID score for 2D editing(e4e, HFGI) is about 0.8, and for structured 3D-aware editing(PREIM3D, IDE-3D) is 0.5~0.6. We have added quantitative results of InstructPix2Pix and img2img on 30 attribute editings, highlighted in red, in Table 5. **Tables 2, 3, and 5 show that our ID score is above 0.6, higher than the previous work. Furthermore, the results of our user study (Table 6) support the notion that our method achieves superior identity preservation and attribute disentanglement.**
>
> We also recognize that there is room for improvement in preserving finer details such as eye shape and eyelashes, and we acknowledge these as areas for future research in this domain.
>
> Moreover, note that the ID score reported in our paper is multi-view identity consistency, which is calculated between the novel views after editing and the input view.
> Like EG3D, the state-of-the-art 3D face generation method, we also uniformly rendered novel views from yaw angles between $[-30^{\\circ},30^{\\circ}]$ and pitch angles between $[-20^{\\circ},20^{\\circ}]$ for an input image.
> EG3D reported a score of 0.77 for identity similarity in synthetic faces in their paper.
> Although we are working with real-world faces, not synthetic ones, and with editing, we still achieve a score of 0.6 or above.
> **To the best of our knowledge, our method is the current state-of-the-art in preserving identity in 3D-aware face editing.
> Therefore, We highlight our ability to preserve identity after editing in different poses.**
>
> To be continued.

---

> ### Author Response · Authors · 2023-11-19
> **Official Response to Reviewer 1rfe (2/2)**
>
> Continued to responses part 1.
>
> > *W3: The proposed method adopts two streams of inputs (real and edited images). However, the ablation study does not show the necessity of them. *
>
> **A**: We appreciate your interest in the necessity of the two streams of inputs (triplet data mechanism) in our method.
> The triplet data mechanism plays a crucial role in achieving accurate and disentangled image editing. To provide a more thorough understanding of its importance, we conducted an ablation study and presented the results below.
>
> Our comparison involves the img2img model, which can be considered as using a text-image pairing data mechanism, in contrast to our method which utilizes the triplet data mechanism. Img2img is trained on the instructions and edited images in our prepared dataset.Unlike InstructPix2NeRF, img2img has no paired images, but the rest of the network structure is same.
>
> **The results of the ablation study, summarized in Table R3.2, demonstrate that the triplet data mechanism significantly contributes to the quality of editing in terms of identity preservation (ID) and attribute dependency (AD) when attribute altering (AA) is close to equal.**
>
> - **ID Score**: Our method consistently outperforms img2img in preserving identity across various attributes, as indicated by the higher ID scores.
>
> - **AA Score**: Our method achieves an editing effect that is close to or superior to that of img2img.
>
> - **AD Score**: The triplet data mechanism helps reduce attribute dependency, ensuring that changes to one attribute do not excessively affect others.
>
> **These results highlight that the triplet data mechanism encourages the model to learn the correlations between changes in pairs of images and the corresponding instructions, leading to more precise and disentangled editing.** In conclusion, the triplet data mechanism is essential for achieving high-quality image editing results.
>
> Table R3.2. The triplet data mechanism ablation. Identity consistency (ID)  measures if it is the same person. Attribute altering (AA) measures the change of the desired attribute. Attribute dependency (AD) measures the change in other attributes when modifying one attribute.
> |Method|bang|eyeglasses|smile|multiple instructions|five o’clock shadow |bushy eyebrows|chubby|double chin|high cheekbones|pale skin|goatee|
> |:--------|:-------:|:-------:|:------:|:------:|:-------:|:-------:|:------:|:------:|:------:|:------:|:------:|
> |img2img-ID$\\uparrow$ |0.40|0.42|0.46|0.37|0.52|0.53|0.50|0.51|0.51|52|0.50|
> |ours-ID$\\uparrow$    |**0.56**|**0.59**|**0.60**|**0.55**|**0.59**|**0.64**|**0.64**|**0.64**|**0.64**|**0.66**|**0.65**|
> |img2img-AA$\\uparrow$ |0.99|3.33|1.47|1.39|0.94|**1.02**|**1.05**|0.96|0.96|1.03|1.12|
> |ours-AA $\\uparrow$  |**1.05**|**3.37**|**1.50**|**1.50**|**0.95**|1.00|**1.05**|**1.00**|**1.07**|**1.11**|**1.21**|
> |img2img-AD$\\downarrow$|0.61|0.79|0.76|0.88|0.77|0.62|0.65|0.76|0.62|0.53|0.57|
> |ours-AD$\\downarrow$   |**0.53**|**0.64**|**0.61**|**0.69**|**0.74**|**0.51**|**0.54**|**0.63**|**0.54**|**0.47**|**0.50**|
>
> ### Summary:
> Thanks for your encouraging review and valuable suggestions. We hope that our added examples and experiments will address your concerns. Moreover, We believe our method will become a strong baseline for future works towards instructed 3D-aware face editing from a single image.

---

> ### Author Response · Authors · 2023-11-22
> **Please let us know if our responses address your concerns.**
>
> Dear reviewer,
>
> We sincerely appreciate the demands on your time, especially during this busy period. After carefully considering your valuable feedback, We have made some new experiments and necessary modifications to our paper. We are wondering if our responses and revision have addressed your concerns.
>
> We are extremely grateful for your time and effort in reviewing our paper, and we sincerely appreciate your feedback. We believe your comments have been instrumental in enhancing the quality of our paper. As today is the last day of the discussion stage, we are kindly awaiting your response.
>
> If you have any further questions or require any additional information from us, please let us know. We would like to invite you to consider our responses and look forward to your reply.
>
> Once again, thank you for your attention and support.
>
> Best regards,
> The Authors

---

> > ### Comment · Reviewer_1rfe · 2023-11-22
> > **Re: Rebuttal**
> >
> > Thanks for the response. I’ve increased my rating.

---

> > > ### Author Response · Authors · 2023-11-22
> > > **Thank you for your review！**
> > >
> > > We are glad that our responses helped and would like to thank you for raising the score of our paper.  Your valuable suggestions are instrumental in making our work more comprehensive.

---

### Official Review · Reviewer_e7JS · 2023-10-31

**Soundness:** 3 good
**Presentation:** 2 fair
**Contribution:** 3 good
**Rating:** 8
**Confidence:** 3

**Summary:**

This paper proposes a method that enables the creation of 3D portraits that have been edited based on text prompts. Leveraging the latent space of EG3D to impose 3D consistency, the proposed method finds a latent vector in the W+ space that matches the edits specified by the prompt and the identity in the input image. A diffusion model conditioned on the input 2D image and the editing prompt is used to predict this latent vector. Additionally, the paper proposes the following 1) Token position randomization to improve the quality of multi-instruction editing 2) An identity consistency module to improve identity preservation during edits.

**Strengths:**

1) While each individual component of the method isn’t novel, the whole method itself is

2) Qualitative results in both the paper and appendix demonstrate plausible editing, though some identity loss remains

3) Quantitative results demonstrate that the method better preserves the identity across edits. The user study additionally bolsters the main contribution of the paper.

**Weaknesses:**

1) The methods section could be written better, with a clear exposition of losses during training and the forward pass during inference. To that end, Fig 2 should be expanded to include both training and inference settings.

2) While the identity consistency is better preserved that prior work, the still remains and identity drift during editing.

**Questions:**

1) Instead of an Encoder, if direct optimization of the W+ vector was used (assuming much larger compute), would it preserve the identity better? What if this is only done during inference and not training?

---

> ### Author Response · Authors · 2023-11-19
> **Official Response to Reviewer e7JS (1/2)**
>
> We sincerely appreciate your interest in our work and your valuable feedback. Below, we provide detailed responses to your comments and questions, addressing your concerns and providing further insights.
>
>
> ### Summary of your key concerns and our improvements:
> 1. You suggest we clarify the exposition of losses during training and the forward pass during inference. (W1)
> - **Response:** We have considered your feedback, made improvements to clarify the total training loss and the forward pass during inference, and modified Figure 2.
> 2. You have some concerns about the preservation of identity. (W2)
> - **Response:** We have added more examples and comparative experiments and show the significant improvement of our method over previous methods through quantitative metrics and user study.
> 3. You are curious about the potential benefits of directly optimizing the $\\mathcal{W+}$ vector for identity preservation. (Q1)
> - **Response:** We conducted experiments to explore this scenario and provide insights into its causes in Appendix A.2.5.
>
> ### Detailed responses to your concerns item-by-item.
>
> > *W1: The methods section could be written better, with a clear exposition of losses during training and the forward pass during inference.*
>
> **A**: We apologize for any confusion regarding our methods section. We have taken your feedback into account and have made improvements to clarify the total training loss and the forward pass during inference.
>
> In Section 4.3, we have included a clear exposition of the total loss used in our method, which consists of the diffusion loss $\\mathcal{L}\_{diff}$ and the identity loss $\\mathcal{L}\_{ID}$. The total loss is expressed as:
> $$
> \\mathcal{L} = \\mathcal{L}\_{diff} + \\lambda\_{id} \\mathcal{L}\_{ID}
> $$
> where $\\lambda\_{id}$ is set to 0.1 in our experiments. This addition aims to provide a more comprehensive understanding of our training process.
>
> **In Section 4.4, we illustrate the inference process in detail.** Specifically, we add a little bit of noise to the latent of the input image (usually 15 steps) to obtain $w\_{ot}$ and then use our model to perform conditional denoising. The model predicts three score estimates, the image-text conditional $\\epsilon\_{\\theta}(w\_{ot}, c\_{I}, c\_{T})$, the only-image conditional $\\epsilon\_{\\theta}(w\_{ot}, c\_{I}, \\emptyset)$, and the unconditional $\\epsilon\_{\\theta}(w\_{ot}, \\emptyset, \\emptyset)$.
> $c\_{T}=\\emptyset$ indicates that the text takes an empty character. $c\_{I}=\\emptyset$ means that the concatenation $w\_{o}$ takes zero and identity modulation takes zero.
> Image and text conditioning sampling can be performed as follows:
> $$
>   \\tilde{\\epsilon}\_{\\theta}(w\_{ot}, c\_{I}, c\_{T})=\\epsilon\_{\\theta}(w\_{ot}, \\emptyset, \\emptyset) + s\_{I}(\\epsilon\_{\\theta}(w\_{ot}, c\_{I}, \\emptyset) - \\epsilon\_{\\theta}(w\_{ot}, \\emptyset, \\emptyset)) + \\\\ s\_{T}(\\epsilon\_{\\theta}(w\_{ot}, c\_{I}, c\_{T}) - \\epsilon\_{\\theta}(w\_{ot}, c\_{I}, \\emptyset))
> $$
> where $s\_{I}$ and $s\_{T}$ are the guidance scales for alignment with the input image and the text instruction, respectively.
>
> To further enhance clarity, **we have also modified Figure 2 to include a switch, distinguishing between the training and inference processes.** This change helps illustrate the transition from training to inference and highlights the specific steps involved.
>
>
> > *W2: While the identity consistency is better preserved than prior work, there still remains identity drift during editing.*
>
> **A**: Thank you for acknowledging our work. Identity preservation and attribute disentanglement have posed challenges for face editing. 3D-aware editing becomes more difficult than 2D editing in this regard since only one input image is visible.
>
> The ID score for 2D editing(e4e, HFGI) is about 0.8, and for structured 3D-aware editing(PREIM3D, IDE-3D) is 0.5~0.6. We have added quantitative results of InstructPix2Pix and img2img on 30 attribute editings, highlighted in red, in Table 5. **Tables 2, 3, and 5 show that our ID score is above 0.6, higher than the previous work. Furthermore, the results of our user study (Table 6) support the notion that our method achieves superior identity preservation and attribute disentanglement.**
>
> We also recognize that there is room for improvement in preserving finer details such as eye shape and eyelashes, and we acknowledge these as areas for future research in this domain.
>
> To be continued.

---

> ### Author Response · Authors · 2023-11-19
> **Official Response to Reviewer e7JS (2/2)**
>
> Continued to responses part 1.
> > *W2: While the identity consistency is better preserved than prior work, there still remains identity drift during editing.* (continued)
>
> Moreover, note that the ID score reported in our paper is multi-view identity consistency, which is calculated between the novel views after editing and the input view.
> Like EG3D, the state-of-the-art 3D face generation method, we also uniformly rendered novel views from yaw angles between $[-30^{\\circ},30^{\\circ}]$ and pitch angles between $[-20^{\\circ},20^{\\circ}]$ for an input image.
> EG3D reported a score of 0.77 for identity similarity in synthetic faces in their paper.
> Although we are working with real-world faces, not synthetic ones, and with editing, we still achieve a score of 0.6 or above.
> **To the best of our knowledge, our method is the current state-of-the-art in preserving identity in 3D-aware face editing.
> Therefore, We highlight our ability to preserve identity after editing in different poses.**
>
> > *Q1: Instead of an Encoder, if direct optimization of the $\\mathcal{W+}$ vector was used (assuming much larger compute), would it preserve the identity better? What if this is only done during inference and not training?*
>
> **A**: Your question is indeed intriguing, and we appreciate your curiosity regarding the potential benefits of direct optimization of the $\\mathcal{W+}$ vector for identity preservation, especially if conducted during inference. We conducted experiments to explore this scenario and provide insights into its causes.
>
> **In our experiments, we considered two configurations: Direct $\\mathcal{W+}$ optimization and PTI optimization.** Direct $\\mathcal{W+}$ optimization involves optimizing the $\\mathcal{W+}$ vector while keeping the generator fixed. PTI (Pivotal Tuning Inversion) technique fine-tunes the generator based on the initial value provided by direct optimization. We conducted 500 steps of optimization on the $\\mathcal{W+}$ vector, and PTI added 100 steps of fine-tuning the generator.
>
> The results of these experiments are presented in Figure 10, where we compare the outcomes of direct $\\mathcal{W+}$ optimization, PTI, and the encoder-based method. The results show that directly replacing the encoder with an optimization method during inference will lead to a severe decrease in both editing effect and identity consistency.
>
> **We attribute this issue to the deviation between the model and data distribution.** The model learns a conditional distribution within the encoder's inversion space during training. When the encoder is replaced by an optimization method during inference, the data distribution used for inference mismatches the learned model distribution. This mismatch results in greater identity drift and undesirable editing outcomes.
>
> While conducting $\\mathcal{W+}$ optimization during training (much larger compute) could potentially address the distribution deviation problem, it may introduce artifacts in novel views, as pointed out by PREIM3D. This is due to optimization being performed on a single image during training.
>
> In summary, while direct optimization of the $\\mathcal{W+}$ vector is an interesting concept, our experiments suggest that it may not necessarily lead to improved identity preservation and editing results compared to the encoder-based approach.
>
> ### Summary:
> Thank you for your encouraging comments and writing suggestions. They are very helpful for us to improve the article. Moreover, we provided new examples and experiments, which we believe can strengthen the breadth and depth of our validation.

---

> ### Author Response · Authors · 2023-11-22
> **Please let us know if our responses address your concerns.**
>
> Dear reviewer,
>
> We sincerely appreciate the demands on your time, especially during this busy period. After carefully considering your valuable feedback, We have made some new experiments and necessary modifications to our paper. We are wondering if our responses and revision have addressed your concerns.
>
> We are extremely grateful for your time and effort in reviewing our paper, and we sincerely appreciate your feedback. We believe your comments have been instrumental in enhancing the quality of our paper. As today is the last day of the discussion stage, we are kindly awaiting your response.
>
> If you have any further questions or require any additional information from us, please let us know. We would like to invite you to consider our responses and look forward to your reply.
>
> Once again, thank you for your attention and support.
>
> Best regards,
> The Authors

---

> > ### Comment · Reviewer_e7JS · 2023-11-22
> > **Rebuttal Response**
> >
> > I would like to thank the authors for the rebuttal. Looking at the rebuttal and other reviews, I have decided to revise my initial rating.

---

> > > ### Author Response · Authors · 2023-11-22
> > > **Thank you for your engagement!**
> > >
> > > We are delighted to have addressed your concerns and would like to express our gratitude for your recognition of our revised version. Your thoughtful and constructive feedback has been integral to making these improvements possible.

---

### Official Review · Reviewer_TVmA · 2023-10-31

**Soundness:** 2 fair
**Presentation:** 2 fair
**Contribution:** 2 fair
**Rating:** 5
**Confidence:** 5

**Summary:**

The proposed approach, InstructPix2NeRF, is an end-to-end model designed for 3D-aware human head editing using a single image and an instructive prompt as inputs. To achieve this results, firstly the authors construct a multimodal 2D human head dataset by leveraging pretrained diffusion models such as e4e and InstructPix2Pix. Secondly, they propose a token position randomization strategy to enhance the model's ability to edit multiple attributes simultaneously. Last, an identity consistency module is incorporated to extract facial identity signals from the input image and guide the editing process. Experimental results demonstrate the effectiveness and superiority of the method.

**Strengths:**

The strengths of the proposed paper can be summarized as:
1. The authors propose a token randomization strategy that can increase the model's capability for editing multiple attributes simultaneously.
2. An identity-preserving module is proposed to guide the editing process and present the original identity in the final outcomes.
3. The proposed method is reported to be time-friendly, producing the results in few seconds.

**Weaknesses:**

The weaknesses of the proposed method can be summarized as:
1. Through the visualization in Figure 1, I find that the original identity and RGB image attributes are not well preserved. Large differences can still be observed in the areas that are not supposed to be edited.
2. Qualitative comparisons. (1) The proposed method seems to struggle with expression editing, e.g., it fails to make the head smiling; The instruct-pix2pix model doesn't encounter this problem; (2) Regarding the "bangs" example, I would prefer the instruct-pix2pix as it contains real bangs; (3) There is no comparisons with Instruct-NeRF2NeRF, AvatarStudio, and HeadSculpt, considering they are more similar works than the compared Talk-to-Edit and img2img; (4) More examples and more scenarios will largely improve the validation. Currently, there are only three types presented.
3. Quantitative comparisons. (1) The evaluations are not comprehensive. Still, only three examples are presented; (2) More quantitative evaluations, e.g., user studies, would be beneficial.

**Questions:**

Besides the weaknesses above, I may have some questions that hope the authors can answer:
1. There lack the reason for generating and using 20-30 instruction prompts for one single paired image. Will the number of instructions affect the training?
2. How will the model perform when it deals with novel characters as in the movie, long hair examples, black men/women, and human of different ages?
3. Will the background affect the edited results? It would be interesting to see the outcomes obtained when editing the same subject against various backgrounds.

---

> ### Author Response · Authors · 2023-11-19
> **Official Response to Reviewer TVmA (1/4)**
>
> Thank you for your detailed review and constructive suggestions. Based on your suggestions, We have improved our paper and a revised version is available now. First, we summarize what we have done this week to address your concerns and provide detailed responses for each weakness and question.
>
> ### Summary of your key concerns and our improvements:
> 1. You have some concerns about the preservation of identities and attributes, as well as the effects of editing. (W1, W2(1), W2(2))
> - **Response:** We have added more examples and comparative experiments and show the significant improvement of our method over previous methods through quantitative metrics and user study.
> 2. You suggest comparing our method to Instruct-NeRF2NeRF, AvatarStudio, and HeadSculpt. (W2(3))
> - **Response:** We discussed and cited these methods in Section 2 and pointed out that they are different from our task and cannot handle 3D-aware editing from a single image.
> 3. You suggest we provide a wider range of examples (W2(4), Q2)
> - **Response:** We provided additional examples and scenarios featuring different races, ages, and hairstyles and discussed the diversity and generalization of our method in Appendix A.3.
> 4. You suggest we conduct a more comprehensive and quantitative evaluation. (W3)
> - **Response:** We conducted a new experiment to compare our method against InstructPix2Pix and img2img on 30 attribute editing types.
> 5. You suggest we discuss the effect of background on editing results. (Q3)
> - **Response:** We conducted a new experiment where we edited images of the same subject in different backgrounds in Appendix A.2.6.
>
> ### Detailed responses to your concerns item-by-item.
> > *W1: The original identity and RGB image attributes are not well preserved.*
>
> **A**: Thanks. Identity preservation and attribute disentanglement have posed challenges for face editing. 3D-aware editing becomes more difficult than 2D editing in this regard since only one input image is visible.
>
> The ID score for 2D editing(e4e, HFGI) is about 0.8, and for structured 3D-aware editing(PREIM3D, IDE-3D) is 0.5~0.6. We have added quantitative results of InstructPix2Pix and img2img on 30 attribute editings, highlighted in red, in Table 5. **Tables 2, 3, and 5 show that our ID score is above 0.6, higher than the previous work. Additionally, the results of our user study (Table 6) support the notion that our method achieves superior identity preservation and attribute disentanglement.**
>
> Furthermore, we introduced attribute dependency (AD) score to measure the change in other attributes when editing certain attributes. **Figure 3 shows that our method preserves the other attributes better when meeting the editing requirements.** For example, in the third row, first column of Figure 3, InstructPix2Pix alters the expression when adding bangs, while our method maintains expression consistency. InstructPix2Pix and img2img often lead to undesirable changes in skin color.
> We acknowledge that there are some limitations in preserving fine details such as eye shape and eyelashes, and we consider these areas for future improvement.
>
> Moreover, note that the ID score reported in our paper is multi-view identity consistency, which is calculated between the novel views after editing and the input view.
> Like EG3D, the state-of-the-art 3D face generation method, we also uniformly rendered novel views from yaw angles between $[-30^{\\circ},30^{\\circ}]$ and pitch angles between $[-20^{\\circ},20^{\\circ}]$ for an input image.
> EG3D reported a score of 0.77 for identity similarity in synthetic faces in their paper.
> Although we are working with real-world faces, not synthetic ones, and with editing, we still achieve a score of 0.6 or above.
> **To the best of our knowledge, our method is the current state-of-the-art in preserving identity in 3D-aware face editing.
> Therefore, We highlight our ability to preserve identity after editing in different poses.**
>
> to be continued

---

> ### Author Response · Authors · 2023-11-19
> **Official Response to Reviewer TVmA (2/4)**
>
> Continued to responses part 1.
>
> >*W2 (1) & (2): The proposed method seems to struggle with expression editing, e.g., it fails to make the head smiling. Regarding the "bangs" example, I would prefer the instruct-pix2pix as it contains real bangs*
>
> **A**:
> Regarding expression editing, it's important to note that our method allows for controlled adjustments of attributes, including expressions. In the case you mentioned, the head does exhibit a smile, albeit to a lesser degree. The degree of the smile can be controlled through the guidance scale, which also affects identity preservation and attribute dependency.
> **As shown in Tables 2, 3, and 5, our method has better ID preservation and less impact on other attributes when obtaining the same editing degree.**
> The ID score of about 0.49 and AD score of above 0.7 indicate that InstructPix2Pix is prone to identity drift and other attribute changes when performing editing. The results of the user study (Table 6) also illustrate this point. For example, in the third row, first column of Figure 3, InstructPix2Pix alters the expression when adding bangs, while our method maintains expression consistency.
>
> For smile editing, you can refer to Figure 11 (first row) and Figure 14 (fourth row) in our paper.
> As for the "bangs" example, you can refer to Figure 15 (third row) and Figure 18 (fourth column). It is true that different methods may have their strengths in specific scenarios, and user preferences may vary. We appreciate your input and understand the appeal of InstructPix2Pix's real bangs. Our method aims to provide versatility for a wide range of editing tasks, particularly those requiring attribute preservation and control.
>
> > *W2 (3): There is no comparisons with Instruct-NeRF2NeRF, AvatarStudio, and HeadSculpt.*
>
> **A**: We appreciate your awareness of these methods and discuss and cite these methods in Section 2. **However, it's crucial to note that these methods have different tasks from our method and cannot handle 3D-aware editing from a single image. While these methods take tens of minutes to optimize a single scene, our method generates 3D-aware editing in seconds for different faces.**
> The input of our method is a single image and user instruction, and the output is edited 3D-aware images and geometry.
>
> - **Instruct-NeRF2NeRF works with reconstructed NeRF scenes, taking as input a set of captured images, their camera poses, and calibration data, while our method operates on a single image with user instructions.**
> Instruct-NeRF2NeRF uses InstructPix2Pix to iteratively edit the set of captured images while optimizing the underlying scene.
>
> - **AvatarStudio utilizes dynamic full-head avatars reconstructed from head videos, which is distinct from our single-image and instruction-based input.**
> AvatarStudio uses their proposed view-and-time-aware Score Distillation Sampling (VT-SDS) with a personalized diffusion model to edit the dynamic NeRF.
>
> - HeadSculpt primarily focuses on coarse-to-fine text-to-3D generation and can achieve editing in the fine stage by blending scores predicted by ControlNet-based InstructPix2Pix and landmark-based ControlNet for the editing instruction and the original description. **However, HeadSculpt is only capable of editing the trained scene model generated in the coarse stage, rather than a single input image.**
>
> Due to these differences in tasks, inputs, and methodologies, a direct comparison with these methods may not provide meaningful insights for evaluating our work. We hope this clarification helps to address your concern.
>
> > *W2 (4): More examples and more scenarios will largely improve the validation.*
>
> **A**: Thank you for the suggestion. We have taken steps to enhance the validation of our work by including more examples and scenarios. In our previous version, we provided examples in Appendix A.3, covering attributes such as beards, hair color, vampires, zombies, old people, children, long hair, etc. in Figures 12 and 13.
> Per your suggestion, **we have further expanded our results to include novel characters from movies (Figure 16), Asians (Figure 14 and 15), and black men and women (Figure 14 and 15).** We believe that these additional results strengthen the breadth and depth of our validation, showing the diversity of our method.
>
> To be continued.

---

> ### Author Response · Authors · 2023-11-19
> **Official Response to Reviewer TVmA (3/4)**
>
> Continued to responses part 2.
>
>  >*W3 (1): The evaluations are not comprehensive. Still, only three examples are presented.*
>
> **A**: We appreciate your feedback on the comprehensiveness of our evaluations. To address your concern, we would like to clarify that we indeed provide a comprehensive evaluation of our method, which includes a broader set of attributes beyond the three examples presented in the main text.
>
> In our previous version, we offered quantitative results for a total of 30 attribute editings of our method, as detailed in Table 5. These results provide a comprehensive assessment of our method's performance across a wide range of attributes. per your suggestion, **we have added quantitative results of InstructPix2Pix and img2img on 30 attribute editings, highlighted in red, in Table 5.**
> The results reveal that when the attribute editing degree AA is comparable to or surpasses the baseline, our method exhibits remarkable enhancement in ID and AD metrics.
>
> > *W3 (2): More quantitative evaluations, e.g., user studies.*
>
> **A**: We apologize for any confusion regarding the presentation of quantitative evaluations, particularly the user study. **In our previous version, we conducted a user study to comprehensively evaluate our method's performance. You can find the user study results in Appendix A.2.4 (Table 6), which provides valuable insights into the quality of our edits.**
>
> The user study involved 1,440 votes from 30 volunteers who evaluated the text instruction correspondence and multi-view identity consistency of editing results. Participants were asked to choose the better result between our method and baseline for various attributes and multiple instructions. The results, presented in Table 6, clearly demonstrate that our method outperforms the baselines, further substantiating the effectiveness of our approach.
>
> We appreciate your feedback and have clarified the location of the user study within the paper for your reference. We copy Table 6 here.
>
> Table 6. The result of our user study. The value represents the rate of Ours > others. Multiple instructions indicate editing with the combinations of the above three attributes.
> |Method|bang|eyeglasses|smile|multiple instructions|
> |:--------|:-------:|:-------:|:------:|:------:|
> |Talk-to-Edit|0.742|0.958|0.817|0.875|
> |InstructPix2Pix|0.833|0.667|0.725|0.683|
> |img2img|0.733|0.758|0.750|0.783|
>
>
> > *Q1: Will the number of instructions affect the training?*
>
> **A**: Your question is indeed valuable, and we appreciate your curiosity regarding the impact of the number of instructions on our training process. Up until now, our approach has primarily drawn inspiration from the success of InstructPix2Pix, employing a range of 20-30 instructions during training. This choice was primarily driven by empirical observations and the composition of the InstructPix2Pix dataset.
>
> Specifically, we performed preprocessing steps where we filtered and removed duplicate face instructions from the InstructPix2Pix dataset. This analysis revealed that, for a single attribute, there were often tens of valid instructions available. By leveraging these insights and filtering out semantically non-compliant instructions generated by ChatGPT, we arrived at the use of 20-30 instructions for training.
>
> We acknowledge the importance of exploring the impact of instruction quantity further. To address this, we are currently conducting training experiments with a smaller or bigger number of instructions. We hope to obtain preliminary results in the near future and look forward to sharing these findings.
>
> > *Q2: How will the model perform when it deals with novel characters as in the movie, long hair examples, black men/women, and human of different ages?*
>
> **A**:
> **Our models are designed to handle a wide range of attributes, including novel characters from movies, varied hairstyles, different races, and varying ages.** As outlined in our experimental settings (Appendix A.1.1), our model is trained on the FFHQ dataset, which comprises 70,000 faces of diverse individuals. The evaluation is conducted on a separate dataset, CelebA-HQ.
>
> In Figure 16, we show the results of editing the characters in the movie's background and lighting conditions from this year's movie "Mission: Impossible – Dead Reckoning Part One."
>
> Additionally, we provide more examples to illustrate the diversity of our method. These examples include long hairs (Figure 12, 13), black men and women (Figure 14, 15), olds (first row of Figure 12, last row of Figure 15), and children (first and fourth row of Figure 12, last row of Figure 14).
>
> We believe that these examples underscore the robustness and applicability of our method across a wide range of faces and scenarios.
>
>
> To be continued.

---

> ### Author Response · Authors · 2023-11-19
> **Official Response to Reviewer TVmA (4/4)**
>
> Continued to responses part 3
>
> > *Q3: Will the background affect the edited results? *
>
> **A**: Thank you for raising this interesting question. To investigate this, we conducted experiments where we edited images of the same subject in different backgrounds.
>
> We show the results in Figure 11, where the first two rows are the same subject, the middle two rows are the same subject and the last two rows are the same subject. The results show that the background has no obvious impact on the editing results.
>
> However, note that when editing colors, particularly when the color being edited is close to the background color, there can be some blending between the foreground and background elements.
>
> We hope this addresses your query and provides insights into the background's role in our editing process.
>
> ### Summary:
> Thanks for your valuable questions and suggestions. Your questions are more about the qualitative and Quantitative evaluation. We did a lot of new experiments and tailored the article accordingly. Hope our response can convince you of the performance of our model.

---

> ### Author Response · Authors · 2023-11-22
> **Please let us know if our responses address your concerns.**
>
> Dear reviewer,
>
> We sincerely appreciate the demands on your time, especially during this busy period. After carefully considering your valuable feedback, We have made some new experiments and necessary modifications to our paper. We are wondering if our responses and revision have addressed your concerns.
>
> We are extremely grateful for your time and effort in reviewing our paper, and we sincerely appreciate your feedback. We believe your comments have been instrumental in enhancing the quality of our paper. As today is the last day of the discussion stage, we are kindly awaiting your response.
>
> If you have any further questions or require any additional information from us, please let us know. We would like to invite you to consider our responses and look forward to your reply.
>
> Once again, thank you for your attention and support.
>
> Best regards,
> The Authors

---

> ### Comment · Reviewer_TVmA · 2023-11-23
>
> Thanks for the detailed response from the authors! However, I still find:
>
> 1. Qualitative results.
>
> (a). Preserving the original identity and RGB attributes for the editing task is crucial in my opinion.
>
> (b). The scenario of "having her smile" appears to be quite challenging. Firstly, the edited expression should ideally be more prominent or noticeable. Secondly, I noticed that other editing tasks, such as "adding bangs to the hairstyle," also result in a slight smile. This raises questions about whether these smiling attributes are inherited from the training data rather than intentional edits.
>
> (c). It would be more fair and beneficial to conduct comparisons with current 3D editing methods, even though they may have different input requirements, or different 3D editing setups, rather than solely comparing it to 2D editing methods.
>
> 2. Comprehensiveness of our evaluations.
>
> (a). The quantitative numbers in Table 5 are not significantly better. Qualitative comparisons would be valuable.
>
> (b). Additionally, expanding the user studies to cover more than just four scenarios would largely enhance the comprehensiveness of your evaluations.
>
> 3. Questions.
>
> (a). It seems that the background has an impact on the results, with a white background yielding the best outcomes.
>
> (b). Based on Appendix A.3, which discusses a wider range of examples, the preservation of identity appears to be a significant challenge. This drawback will be pronounced considering identity-preserving is a most important contribution claimed by the authors.
>
> Thanks again! However, overall based on the novelty, contributions, and the authors' response, I am sorry that I would prefer to maintain or reduce my original score.

---

> > ### Author Response · Authors · 2023-11-23
> > **Official Response to Reviewer TVmA (1/2)**
> >
> > Thanks for your valuable questions and constructive suggestions. Below, we provide detailed responses to your comments and questions, addressing your concerns and providing further insights.
> >
> > ### Detailed responses to your concerns item-by-item.
> > 1. Qualitative results
> > > (a). Preserving the original identity and RGB attributes for the editing task is crucial.
> >
> > **A**:
> > Certainly, we fully agree with you that identity preservation and attribute disentanglement are crucial for the face editing task, whether it involves 2D or 3D face editing. **When working with a single image, achieving 3D-aware editing while maintaining 3D consistency can indeed be a challenging task.** This remains an open question that requires continuous improvements. We have taken measures to address this challenge. **Compared to existing methods (Table 2, 3, 5, and 6; Figure 3 and 18) for achieving 3D-aware editing from a single image, our method is superior.**
> >
> > > (b). The scenario of "having her smile" appears to be quite challenging.
> >
> > **A**:
> > We appreciate your astute observation regarding the smile.
> > Regarding the smile, Our method, like previous approaches, relies on guidance scales to manipulate the intensity of the smile. We guess you prefer a bigger smile. **For the results of a bigger smile, you can refer to Figure 11 (first row), Figure 14 (fourth row), and Figure 10 (second row, fourth column) in our paper, which shows more prominent and noticeable smiles**
> >
> > You have also noticed that "adding bangs to the hairstyle" in Figure 3 (first column) also results in a slight smile. We think this is an isolated case.
> > **However, it is clear to see that our method causes far fewer smiles than the baselines. This suggests that our method is better at disentangling attributes, and the metric AD (The degree of other attribute changes when performing editing) in Table 2 supports this statement as well. This is a critical improvement on face editing.**
> >
> > Furthermore, we acknowledge that slight smile changes could be influenced by dataset biases.
> > As StyleGAN[1] reports, the FFHQ was crawled from Flickr, inheriting all the biases of that website.
> > InterFaceGAN [2] investigated the correlation between attributes and found that "Smile" entangles slightly with some other attributes, which is consistent with the conclusion of Karras et al. [1].
> > However, we want to clarify that the examples we have provided in Figure 11 (first row), Figure 14 (fourth row), and Figure 10 (second row, fourth column) in our paper, show that **the degree of smile editing is much more prominent and noticeable than the smile bias. This suggests that the smiles observed in our results are due to the editing operations rather than biases in the dataset.**
> >
> > > (c). It would be more fair and beneficial to conduct comparisons with different 3D editing setups, rather than solely comparing it to 2D editing methods.
> >
> > **A**:
> > We apologize for the confusion about baselines. We will revise the baseline description to make it clearer at the beginning of Section Experiments.
> > **Our chosen baselines are not 2D editing methods. Talk-To-Edit and InstructPix2Pix in our paper are two-stage 3D-aware editing methods, combining 2D text-guided editing and 3D inversion. Img2img is a text-to-3D model-based editing method.**
> > Subsequently, we will try to compare it with the methods you mentioned that require multi-view image or video input.
> >
> >
> > 2. Comprehensiveness of our evaluations.
> > > (a). The quantitative numbers in Table 5 are not significantly better.
> >
> > **A**:
> > We apologize for the inconvenience,  we have listed the metrics on 30 attribute editing types in Table 5, but the average metric is missing.
> > **As shown in Table R1.1, the average metrics indicate that with leading editing effects, our ID and AD are better than the baselines.**
> >
> > Table R1.1 **The average results in Table 5.** Identity consistency (ID)  measures if it is the same person. Attribute altering (AA) measures the change of the desired attribute. Attribute dependency (AD) measures the change in other attributes when modifying one attribute.
> > |Method|ID$_{avg}\uparrow$|AA$_{avg}\uparrow$|AD$_{avg}\downarrow$|
> > |:--------|:-------:|:-------:|:------:|
> > |InstructPix2Pix|0.487|0.494|0.643|
> > |img2img|0.502|0.678|0.643|
> > |InstructPix2NeRF|**0.615**|**0.831**|**0.574**|
> >
> > Moreover, to assess human opinion on the visual editing results, we conduct a user study which contributes to the qualitative comparisons in Table 6.
> > We provide some examples from the user study in Figure 18.
> >
> > To be continued.

---

> > ### Author Response · Authors · 2023-11-23
> > **Official Response to Reviewer TVmA (2/2)**
> >
> > Continued to response part 1
> >
> > > (b). Expanding the user studies to cover more than just four scenarios would largely enhance the comprehensiveness of your evaluations.
> >
> > **A**:
> > Thank you for the suggestion.
> > For the qualitative evluation, user study plays an important role, and more scenarios will lead to a more comprehensive evaluation.
> > Regarding the number of attributes, we follow the previous face editing methods e4e[3], HFGI[4], Style Transformer[5], and PREIM3D[6].
> >
> > Per on your suggestions, we will add some scenarios for the user study in the next revision. So, we will recruit more volunteers and spend some more money in order to provide enough votes for each scenario.
> >
> >
> > 3. Questions.
> > > (a). It seems that the background has an impact on the results, with a white background yielding the best outcomes.
> >
> > Thank you very much for this interesting observation, pointing to a meaningful direction of research. We will conduct a more comprehensive study in our future work.
> >
> >
> > > (b). Identity preservation appears to be a significant challenge.
> >
> > Thank you for the comment,
> > Identity preservation is a challenge in the field of face editing especially in 3D-aware face editing since only one input image is visible. We have made efforts to mitigate this problem.
> > At present, our method represents an advancement over other methods.
> > This is still an open problem, and we hope the community to work together to solve it.
> >
> > By the way, we calculated the ID similarity for some identical and different subjects in the CelebA dataset.
> > The average ID similarity is 0.48 for the same subject and 0.12 for different subjects.
> >
> > To the best of my knowledge, we are the first to solve instructed 3D-aware portrait editing from a single real image. To solve the problem, we prepare a triplet dataset for the human face domain and propose an end-to-end framework InstructPix2NeRF with identity consistency module and token position randomization training strategy.
> >
> > We will continue to make efforts to improve identity preservation and attribute disentanglement.
> >
> >
> > [1] Karras, Tero, Samuli Laine, and Timo Aila. "A style-based generator architecture for generative adversarial networks." Proceedings of the IEEE/CVF conference on computer vision and pattern recognition. 2019.
> >
> > [2] Shen, Yujun, et al. "Interfacegan: Interpreting the disentangled face representation learned by gans." IEEE transactions on pattern analysis and machine intelligence 44.4 (2020): 2004-2018.
> >
> > [3] Tov, Omer, et al. "Designing an encoder for stylegan image manipulation." ACM Transactions on Graphics (TOG) 40.4 (2021): 1-14.
> >
> > [4] Wang, Tengfei, et al. "High-fidelity gan inversion for image attribute editing." Proceedings of the IEEE/CVF Conference on Computer Vision and Pattern Recognition. 2022.
> >
> > [5] Hu, Xueqi, et al. "Style transformer for image inversion and editing." Proceedings of the IEEE/CVF Conference on Computer Vision and Pattern Recognition. 2022.
> >
> > [6] Li, Jianhui, et al. "PREIM3D: 3D Consistent Precise Image Attribute Editing from a Single Image." Proceedings of the IEEE/CVF Conference on Computer Vision and Pattern Recognition. 2023.
> >
> > ### Summary
> > Once again thanks for your valuable questions and suggestions. We hope our response can address your concerns. Your valuable suggestions are instrumental in making our work more comprehensive.

---

> ### Comment · Reviewer_TVmA · 2023-12-03
>
> Following my previous responses and the authors' reply:
> 1. It's still unclear how the authors build their baselines for experiments. There is no explanation as to why they don't build the baselines based on Instruct-NeRF2NeRF, AvatarStudio, or HeadSculpt, now that they have built some baselines on their own.
> 2. I find it hard to agree with the authors' comments that their results are superior to their compared methods. I would say that "slightly better" or "a new editing task" would be more reasonable phrases.
> 3. I still have concerns regarding the "smiling issue". While the authors comment that the smiling issue is dependent on the case and guidance scale, the guidance scales they used for different cases and different methods are not provided. Additionally, solving the "smiling issue" to some extent rather than addressing it comprehensively across all cases doesn't seem to be a reasonable result for a paper in a top-venue conference like ICLR.
>
> Based on my initial reviews and following discussions, I am sorry that I still recommend rejection.

---

### Author Response · Authors · 2023-11-19
**Summary of Paper Revision**

We sincerely thank all reviewers for their valuable comments and remarkably insightful suggestions. In this revised version, we have diligently addressed each of the reviewer's comments and incorporated their valuable input. All reviserd parts are highlighted in red. Here we summarize changes to the revised version.

### Summary of paper revisions
**Experiments**
- We discuss the diversity and generalization of our method and provide more examples and scenarios involving novel characters of movies, Asians, black men and women, old people, children, and long hair examples in Appendix A.3. (Reviewer 1rfe, Reviewer TVmA)
- We explore the performance of replacing the encoder with optimization of W+ during inference in Appendix A.2.5. (Reviewer e7JS)
- We present the results when editing the same subject against various backgrounds in Appendix A.2.6. (Reviewer TVmA)
- We add quantitative comparisons with instruct-pix2pix and img2img on more editing types in Appendix A.2.3. (Reviewer TVmA)
- We add the triplet data mechanism ablation in Appendix A.2.7. (Reviewer 1rfe)

**Writing**
- We have clarified the losses during training and the forward pass during inference in Section 4 and modified Figure 2. (Reviewer e7JS)
- We have discussed the related works, Instruct-NeRF2NeRF, AvatarStudio, and HeadSculpt in Section 2. (Reviewer TVmA)
- We have analyzed limitations in preserving fine details in Section 6. (Reviewer 1rfe)
- We have revised some typos.

For a detailed breakdown of these revisions, we refer you to the official comments.
We also address the questions and concerns of the reviewers in detail and would like to clarify any further concerns. We are open to further improving our work in all aspects.

---

### Meta-Review · Area_Chair_Dugr · 2023-12-07

**Metareview:**

The paper proposes a new task that enables instructed 3D-aware portrait editing from a single image. The paper received two positive ratings and one negative ratings. The positive reviews recognize the novel task setting proposed in paper. The negative reviews mostly concerned the visual quality, especially the identity consistency preservation. As the task is novel and meaningful, the proposed method and experiment validation is reasonable. We would like to recommend its acceptance. To further strength the paper, we would recommend the authors to conduct additional comparisons to those straightforward A+B baselines that can potentially do (or show that they cannot do) the jobs. For example, instruction + EG3D/IDE-3D, instructpix2pix on 2D + 3D lifting, etc.

**Justification For Why Not Higher Score:**

Important baseline comparisons missing. Visual quality is not ideal.

**Justification For Why Not Lower Score:**

The proposed task is novel and has interesting applications.

---

### Decision · Program_Chairs · 2024-01-16

Accept (poster)